# Unity by Diversity: Improved Representation Learning for Multimodal VAEs

**Thomas M. Sutter**[1,*] **Yang Meng**[3], **Andrea Agostini**[1], **Daphné Chopard**[1,2],
**Norbert Fortin**[4], **Julia E. Vogt**[1], **Babak Shahbaba**[3], **Stephan Mandt**[3,5]

[1]Department of Computer Science, ETH Zurich
[2]Department of Intensive Care and Neonatology, University Children's Hospital Zurich
[3]Department of Statistics, UC Irvine
[4]Department of Neurobiology and Behavior, UC Irvine
[5]Department of Computer Science, UC Irvine

## Abstract

Variational Autoencoders for multimodal data hold promise for many tasks in data analysis, such as representation learning, conditional generation, and imputation. Current architectures either share the encoder output, decoder input, or both across modalities to learn a shared representation. Such architectures impose hard constraints on the model. In this work, we show that a better latent representation can be obtained by replacing these hard constraints with a soft constraint. We propose a new mixture-of-experts prior, softly guiding each modality's latent representation towards a shared aggregate posterior. This approach results in a superior latent representation and allows each encoding to preserve information better from its uncompressed original features. In extensive experiments on multiple benchmark datasets and two challenging real-world datasets, we show improved learned latent representations and imputation of missing data modalities compared to existing methods.

## 1 Introduction

The fusion of diverse modalities and data types is transforming our understanding of complex phenomena, enabling more nuanced and comprehensive insights through the integration of varied information sources. Consider, for instance, the role of a medical practitioner who synthesizes multiple tests and measurements during diagnosis and treatment. This process involves merging shared information across different tests and identifying test-specific details, both of which are critical for optimal patient care and medical decision-making.

Among the existing methods, multimodal Variational Autoencoders (VAEs) have emerged as a promising approach for jointly modeling and learning from weakly-supervised heterogeneous data sources. While scalable multimodal VAEs utilizing a single shared latent space efficiently handle multiple modalities [Wu and Goodman, 2018, Shi et al., 2019, Sutter et al., 2021], finding an optimal method to aggregate these modalities remains challenging. The aggregation methods and resulting joint representations are often suboptimal and overly restrictive [Daunhawer et al., 2022, Sutter, 2023], leading to inferior latent representations and generative quality. This trade-off between shared and modality-specific information in the latent representations of multimodal VAEs results in limited quality or coherence in generated samples, even in relatively simple scenarios.

---

*Corresponding author: `thomas.sutter@inf.ethz.ch`

38th Conference on Neural Information Processing Systems (NeurIPS 2024).

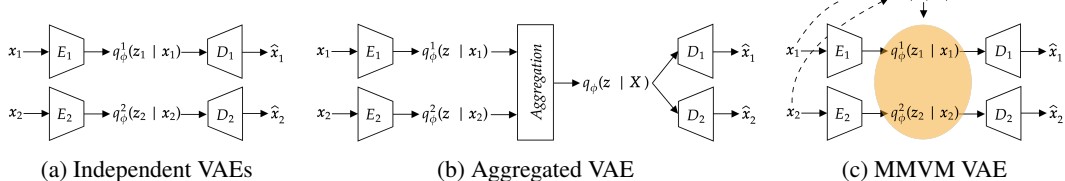

|  |  |  |
|:-:|:-:|:-:|
| (a) Independent VAEs | (b) Aggregated VAE | (c) MMVM VAE |

Figure 1: Independent VAEs (Figure 1a) provide reconstructions for individual modalities but lack information sharing across modalities. Multimodal VAEs with joint posterior approximation (Figure 1b) aggregate unimodal posteriors into a joint posterior but may incur poor reconstruction quality. Our proposed MMVM VAE (Figure 1c) enhances independent VAEs with a data-dependent prior, $h(\boldsymbol{z} \mid \boldsymbol{X})$, allowing soft-sharing of information between modalities while preserving modality-specific reconstructions.

In this work, we propose a novel multimodal VAE, termed the multimodal variational mixture-of-experts prior (MMVM) VAE, to overcome the aforementioned limitations. Instead of modeling the dependencies between different modalities through a joint posterior approximation, we introduce a multimodal and data-dependent prior distribution (see Figure 1). Our proposed multimodal objective is inspired by the VAMP-prior formulation introduced by Tomczak and Welling [2017], which is traditionally used to learn an optimal prior distribution between unimodal data samples, whereas we aim for an optimal prior between different modalities of the same data sample. The resulting regularization term in the VAE objective can be interpreted as minimizing the distance between positive pairs, similar to contrastive learning methods [Oord et al., 2019, Tian et al., 2020]; see Section 4 for details.

We demonstrate the superior performance of the MMVM VAE on three multimodal benchmark datasets, comparing it to unimodal VAEs and multimodal VAEs with joint posterior approximations. Our evaluation focuses on the generative coherence and the quality of the learned latent representations. While independent unimodal VAEs fail to leverage additional modalities during training, they avoid multimodal aggregation disturbances in data reconstruction. On the other hand, multimodal VAEs with a joint posterior approximation must combine both shared and modality-specific information. Previous work by Daunhawer et al. [2022] has shown that this approach results in a trade-off between reconstruction quality and learned latent representation. In contrast, the MMVM VAE accurately reconstructs all modalities and learns meaningful latent representations.

In more practical settings, we address two challenging tasks from the neuroscience and medical domain. First, we analyze hippocampal neural activities from multiple subjects in a memory experiment [Allen et al., 2016]. By treating each subject as a modality, our MMVM VAE enables the description of underlying neural patterns shared across subjects while quantifying individual differences in brain activity and behavior, thereby providing potential insights into the neural mechanisms underlying memory impairment. Second, we tackle identifying cardiopulmonary diseases from chest radiographs using the MIMIC-CXR dataset [Johnson et al., 2019b], which reflects real-world conditions with images of varying quality. By leveraging both frontal and lateral X-ray views as distinct modalities, our MMVM method learns representations that consistently improve disease classification compared to existing VAEs.

This paper advances multimodal machine learning by providing a robust framework for integrating diverse data types and improving the quality of learned representations and generative models.

## 2   Related Work

**Multimodal Learning.** While there is a long line of research on multimodal machine learning (ML) [Baltrušaitis et al., 2018, Liang et al., 2022], multimodal generative ML has gained additional attraction in recent years [Manduchi et al., 2024], driven by impressive results in text-to-image generation [Ramesh et al., 2021, 2022, Saharia et al., 2022]. Unlike these methods, we focus on scalable methods that are designed for a large number of modalities to generate any modality from any other modality without having to train a prohibitive number of different models[2].

---

[2]There are $2^M - 1$ different subsets for a dataset of $M$ modalities and, hence, paths for any-to-any mappings.

**Multimodal VAEs.** Scalable multimodal VAEs using a joint posterior approximation are based on aggregation in the latent space[3]. Multimodal VAEs that learn a joint posterior approximation of all modalities [e.g., Wu and Goodman, 2018, Shi et al., 2019, Sutter et al., 2021] require restrictive assumptions, which lead to inferior performance. Daunhawer et al. [2022] show that aggregation-based multimodal VAEs cannot achieve the same generative quality as unimodal VAEs and struggle with learning meaningful representations depending on the relation between modalities. If we can predict one modality from another, mixture-of-experts-based posterior approximations perform best if only a single modality is given as input, while product-of-experts-based approximations excel if the full set of modalities is available. Extensions [Sutter et al., 2020, Daunhawer et al., 2020, Palumbo et al., 2023] have introduced modality-specific latent subspaces that lead to improved generative quality but cannot completely overcome these limitations. In contrast, the proposed MMVM method uses neither an aggregated latent space nor modality-specific latent subspaces as in previous works. It only leverages a data-dependent prior distribution to regularize the learned posterior approximations. A related line of work with different constraints is multiview VAEs [Bouchacourt et al., 2018, Hosoya, 2018]. In contrast to multimodal VAEs, multiview VAEs often use a single encoder and decoder for all views (thereby sharing the parameter weights between views). While initial attempts also assume knowledge about the number of shared and independent generative factors, extensions [Locatello et al., 2020, Sutter et al., 2023a,b] infer these properties during training.

**Role of Prior in VAE Formulations.** Tomczak and Welling [2017] first incorporated data-dependent priors into VAEs by introducing the VAMP-prior. In contrast to Tomczak and Welling [2017], who are primarily interested in better ELBO approximations, our focus is on learning better multimodal representations and overcoming the limitations faced in previous multimodal VAE works. Sutter et al. [2020] used a data-dependent prior combined with a joint posterior approximation defining a Jensen-Shannon divergence regularization based on the geometric mean. However, their work lacks a rigorous derivation and relies on the suboptimal conditional generation during training [Daunhawer et al., 2022]. Joy et al. [2021] also presented a multimodal VAE inspired by the VAMP-prior VAE. They leverage the VAMP prior to model missing modalities rather than using it as a regularization objective between multimodal samples, as we do in this work. An additional line of work [e.g., Bhattacharyya et al., 2019, Mahajan et al., 2020] leverages normalizing flows to increase the expressivity of the multimodal prior distribution, but this sacrifices the method's scalability.

## 3 Background on Multimodal VAEs

**Problem Specification.** We consider a dataset $\mathbb{X} = \{\boldsymbol{X}^{(i)}\}_{i=1}^n$ where each $\boldsymbol{X}^{(i)} = \{\boldsymbol{x}_1^{(i)}, \ldots, \boldsymbol{x}_M^{(i)}\}$ is a set of $M$ modalities $\boldsymbol{x}_m$ with latent variables $\boldsymbol{z} = \{\boldsymbol{z}_1^{(i)}, \ldots, \boldsymbol{z}_M^{(i)}\}$. The modalities $\boldsymbol{x}_m^{(i)}$ could represent images of the same object taken from different camera angles, text-image pairs, or—as in this paper—neuroscience data from different animal subjects with shared experimental conditions and multiple medical measurements of a patient. When contextually clear, we remove the superscript $(i)$ to improve readability.

Inspired by variational autoencoders [VAEs, Kingma and Welling, 2014], we aim to learn an objective for representation learning while sharing information from different data modalities. For example, we would like to embed neuroscience data into a shared latent space to make brain activations comparable across subjects. At the same time, we want to avoid imposing assumptions on information sharing that are too strong to be able to take individual traits of the data modalities into account. As is typical in VAEs, this procedure involves a decoder (or likelihood) $p_\theta(\boldsymbol{X} \mid \boldsymbol{z})$, an encoder (or variational distribution) $q_\phi(\boldsymbol{z} \mid \boldsymbol{X})$, and a prior $h(\boldsymbol{z}|\boldsymbol{X})$ that we allow to depend on the input.

**Data-Dependent Prior and Objective.** The VAE framework allows us to derive an ELBO-like learning objective $\mathcal{E}$ as follows

$$\mathcal{E}(\boldsymbol{X}) = \mathbb{E}_{q_\phi(\boldsymbol{z}|\boldsymbol{X})} \left[ \log p_\theta(\boldsymbol{X} \mid \boldsymbol{z}) - \log \frac{q_\phi(\boldsymbol{z} \mid \boldsymbol{X})}{h(\boldsymbol{z}|\boldsymbol{X})} \right].$$

Above, $\theta$ and $\phi$ denote the learnable model variational parameters. Importantly, our approach allows for an input-dependent prior $h(\boldsymbol{z} \mid \boldsymbol{X})$. Data-dependent priors can be justified from an empirical Bayes standpoint [Efron, 2012] and enable information sharing across data points with

---

[3]Sutter et al. [2021] describe how different implementations of joint posterior multimodal VAEs relate to different abstract mean definitions.

an intrinsic multimodal structure, as in our framework. They effectively amortize computation over many interrelated inference tasks. We stress that by making the prior data dependent, our model no longer allows for unconditional generation; however, this property can be restored by incorporating pseudo inputs [Tomczak and Welling, 2017], hyper-priors [Sønderby et al., 2016], or ex-post density estimation techniques [Ghosh et al., 2019]. We discuss the objective in more detail in Appendix A, where we prove that the resulting objective is upper bounded by the mean squared reconstruction error, ensuring the existence of (local) optima and thus tractable optimization.

**Encoder and Decoder.** We now specify our encoder and decoder assumptions. A simple encoder choice relies on a single neural network encoder that expects multi-modal inputs, but this approach fails if one or more modalities are missing [Suzuki and Matsuo, 2022]. This shortcoming has motivated multiple approaches [Wu and Goodman, 2018, Shi et al., 2019, Sutter et al., 2021] with separate encoders $q_\phi^m(z_m|x_m)$—one for each modality $m$—that are then *aggregated* in the latent space, e.g., by using a product or mixture distribution. Samples drawn from the joint distribution, e.g., $q_\phi(z \mid X) = \frac{1}{M} \sum_{m=1}^M q_\phi^m(z \mid x_m)$, reconstruct all modalities:

$$\mathcal{E}(X) = \mathbb{E}_{q_\phi(z|X)} \left[ \log h(z \mid X) \right] + \mathbb{E}_{q_\phi(z|X)} \left[ \log \frac{p_\theta^m(x_m \mid z)}{q_\phi(z \mid X)} \right]. \tag{1}$$

As argued and discussed in this paper, such aggregation can be overly restrictive. Instead, this paper explores a different aggregation mechanism that preserves the individual encoders *and* decoders for each modality. Hence, we assume independent decoders $p_\theta^m(x_m|z_m)$ for every modality $m$, assuming conditional independence of each modality given their latent representation [see also Wu and Goodman, 2018, Shi et al., 2019, Sutter et al., 2021].

Following this assumption, we rewrite the objective $\mathcal{E}$ as

$$\mathcal{E}(X) = \mathbb{E}_{q_\phi(z|X)} \left[ \log h(z \mid X) \right] + \sum_{m=1}^M \mathbb{E}_{q_\phi(z_m|x_m)} \left[ \log \frac{p_\theta^m(x_m \mid z_m)}{q_\phi^m(z_m \mid x_m)} \right]. \tag{2}$$

Our assumptions imply that the likelihood and posterior entropy terms (the second term in Equation (2)) decouple across modalities, i.e. $q_\phi(z \mid X) = \prod_{m=1}^M q_\phi^m(z_m \mid x_m)$ and $p_\theta(X \mid z) = \prod_{m=1}^M p_\theta(x_m \mid z_m)$. In contrast, the cross-entropy between the encoder and prior (the first term in Equation (2)) does not decouple and may result in information sharing across modalities. We specify further design choices in the next section.

## 4 Multimodal Variational Mixture VAE

We propose the multimodal variational mixture-of-experts prior (MMVM) VAE, a novel multimodal VAE. The main idea is to design a mixture-of-experts prior across modalities that induces a soft-sharing of information between modality-specific latent representations rather than hard-coding this through an aggregation approach.

VAEs are an appealing model class that allows us to infer meaningful representations and preserve modality-specific information due to the reconstruction loss. Contrastive learning approaches, on the other hand, have shown impressive results on representation learning tasks related to extracting shared information between modalities by maximizing the similarity of their representations [Radford et al., 2021]. Contrastive approaches focus on the shared information between modalities, neglecting potentially useful modality-specific information. We are interested in preserving modality-specific information, which is necessary to generate missing modalities conditionally.

Therefore, we leverage the idea of maximizing the similarity of representations for *generative models*. We propose a prior distribution that models the dependency between the different views and a new multimodal objective that encourages similarity between the unimodal posterior approximations $q_\phi^m(z_m \mid x_m)$ using the regularization term in the objective as a "soft-alignment" without the need for an aggregation-based joint posterior approximation. We discuss objectives based on data-dependent priors in more detail in Appendix A.

To this end, we define a data-dependent MMVM prior distribution in the form of a mixture-of-experts distribution of all unimodal posterior approximations

$$h(\boldsymbol{z} \mid \boldsymbol{X}) = \prod_{m=1}^{M} h(\boldsymbol{z}_m \mid \boldsymbol{X}) \quad \text{where} \quad h(\boldsymbol{z}_m \mid \boldsymbol{X}) = \frac{1}{M} \sum_{\tilde{m}=1}^{M} q_\phi^{\tilde{m}}(\boldsymbol{z}_m \mid \boldsymbol{x}_{\tilde{m}}), \quad \forall\, m \leq M. \quad (3)$$

This notation implies that we use the variational distributions of all modalities $\tilde{m}$ to construct a mixture distribution and then use the same mixture distribution as a prior for any modality $m$. Finally, we build the product distribution over the $M$ components.

Our construction of a variational mixture of posteriors is similar to the VAMP-prior of Tomczak and Welling [2017] that proposes the aggregate posterior $q(\boldsymbol{z}) \equiv \frac{1}{N} \sum_{i=1}^{n} q_\phi(\boldsymbol{z} \mid \boldsymbol{x}^{(i)})$ of a unimodal VAE as a prior. Note, however, that our approach considers mixtures in *modality* space and not data space. In contrast to Tomczak and Welling [2017], our variational mixture is conditioned on a specific instance $\boldsymbol{X}$ and, therefore, does not share information across different instances $\boldsymbol{X}^{(i)} \in \mathbb{X}$. Rather, we share information across the different modalities $\boldsymbol{x}_m^{(i)} \in \boldsymbol{X}^{(i)}$ *within* a given instance. Intuitively, we build the *aggregate posterior* in modality space and replicate this aggregate posterior over all modalities. We stress that this aggregate posterior differs from the standard definition as an average of variational posteriors over the empirical data distribution. Even though the prior appears factorized over the modality space, each factor still shares information across all data modalities by conditioning on the multimodal feature vector $\boldsymbol{X}$ (Equation (3)).

Figure 1 graphically illustrates the behavior of the proposed MMVM VAE compared to a set of independent VAEs and an aggregation-based multimodal VAE. A set of independent VAEs (Figure 1a) cannot share information among modalities. Aggregation-based VAEs (Figure 1b), in contrast, enforce a shared representation between the modalities. The MMVM VAE (Figure 1c) enables the soft-sharing of information between modalities through its input data-dependent prior $h(\boldsymbol{z} \mid \boldsymbol{X})$.

**Minimizing Jenson-Shannon Divergence.** The "rate" term $R$ in the objective, i.e., the combination of variational entropy and cross-entropy, reveals a better understanding of the effect of the mixture prior. Defining $R = KL(q_\phi(\boldsymbol{z} \mid \boldsymbol{X}) \| h(\boldsymbol{z}|\boldsymbol{X}))$ where $KL$ denotes the Kullback-Leibler divergence, the factorization in Equation (3) implies that

$$R = \sum_{m=1}^{M} KL\left(q_\phi^m(\boldsymbol{z}_m|\boldsymbol{x}_m) \| \frac{1}{M} \sum_{\tilde{m}}^{M} q_\phi^{\tilde{m}}(\boldsymbol{z}_m|\boldsymbol{x}_{\tilde{m}})\right) = M \cdot JS(q_\phi^1(\boldsymbol{z}_1|\boldsymbol{x}_1), \ldots, q_\phi^M(\boldsymbol{z}_M|\boldsymbol{x}_M)),$$

where $JS(\cdot)$ is the Jensen-Shannon divergence between $M$ distributions [Lin, 1991]. Hence, maximizing the objective $\mathcal{E}(\boldsymbol{X})$ of the proposed MMVM VAE is equal to minimizing $M$ times the JS divergence between all the unimodal posterior approximations $q_\phi^m(\boldsymbol{z}_m \mid \boldsymbol{x}_m)$. Minimizing the Jensen-Shannon divergence between the posterior approximations is directly related to pairwise similarities between posterior approximation distributions of positive pairs, similar to contrastive learning approaches but in a generative approach.

### 4.1   Optimality of the MMVM Prior

Lemma 4.1 shows that Equation (3) is *optimal* in the sense that it is the unique minimizer of the cross entropy between our chosen variational distribution and an arbitrary prior.

**Lemma 4.1.** *The expectation on the right-hand side of Equation (2) is maximized when for each $m \in \{1, \cdots, M\}$, the prior $h(\boldsymbol{z}_m|\boldsymbol{X})$ is equal to the aggregated posterior of a multimodal sample given on the first line of Equation (3).*

*Proof.* Since the cross-entropy term in Equation (2) involves an expectation over the data $\boldsymbol{X}$ and both $q_\phi(\boldsymbol{z} \mid \boldsymbol{X})$ and $h(\boldsymbol{z} \mid \boldsymbol{X})$ depend on $\boldsymbol{X}$, we can prove the identity for a given value of $\boldsymbol{X}$.

We exploit the factorization of both the variational posterior and the prior over the modalities. Interpreting the cross-entropy between the variational distribution and prior as a functional $F$ of the

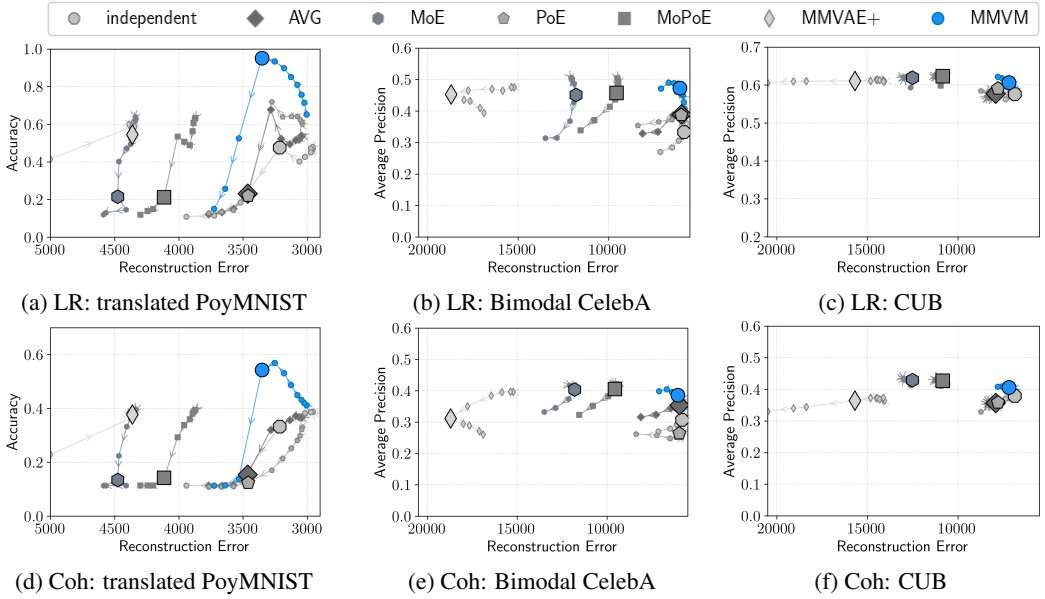

| (a) LR: translated PoyMNIST | (b) LR: Bimodal CelebA | (c) LR: CUB |
| (d) Coh: translated PoyMNIST | (e) Coh: Bimodal CelebA | (f) Coh: CUB |

Figure 2: Results on the benchmark datasets translated PolyMNIST, bimodal CelebA, and CUB. An optimal model would be in the top right corner with low reconstruction error and high classification performance. The proposed MMVM method either achieves a higher classification performance, latent representation (LR, Figures 2a to 2c) or coherence of generated samples (Coh, Figures 2d to 2f), with the same reconstruction loss or the same classification performance with lower reconstruction loss. Every point averages runs over multiple seeds and a specific $\beta$ value (see Section 5.1).

prior $h$, we have

$$F[h(\boldsymbol{z}|\boldsymbol{X}))] \equiv \mathbb{E}_{q_\phi(\boldsymbol{z}|\boldsymbol{X})}\left[\log h(\boldsymbol{z}|\boldsymbol{X}))\right] = \mathbb{E}_{\prod_{m=1}^M q_\phi^m(\boldsymbol{z}_m|\boldsymbol{x}_m)}\left[\log \prod_{m=1}^M h(\boldsymbol{z}_m|\boldsymbol{X}))\right]$$

$$= \sum_{m=1}^M \mathbb{E}_{q_\phi^m(\boldsymbol{z}_m|\boldsymbol{x}_m)}\left[\log h(\boldsymbol{z}_m|\boldsymbol{X}))\right] = M \cdot \mathbb{E}_{\frac{1}{M}\sum_{\tilde{m}=1}^M q_\phi^{\tilde{m}}(\boldsymbol{z}_m|\boldsymbol{x}_{\tilde{m}})}\left[\log h(\boldsymbol{z}_m|\boldsymbol{X}))\right].$$

As a result, we see that $F[h(\cdot)]$ is an expectation over a mixture distribution. We can solve for the optimal distribution $h(\cdot)$ by adding a Lagrange multiplier that enforces $h(\cdot)$ normalizes to one:

$$\max_{h(\boldsymbol{z}_m|\boldsymbol{X})} F[h(\boldsymbol{z}_m \mid \boldsymbol{X})] + \gamma\left(\int h(\boldsymbol{z}_m \mid \boldsymbol{X})d\boldsymbol{z}_m - 1\right) = \max_{h(\boldsymbol{z}_m|\boldsymbol{X})} \mathcal{L}[h,\gamma]$$

To maximize the Lagrange functional $\mathcal{L}[h,\gamma]$, we compute its (functional) derivatives with respect to $h(\boldsymbol{z}_m|\boldsymbol{X})$ and $\gamma$.

$$\frac{\partial \mathcal{L}[h(\boldsymbol{z}_m|\boldsymbol{X}),\gamma]}{\partial h(\boldsymbol{z}_m|\boldsymbol{X})} = \frac{\frac{1}{M}\sum_{\tilde{m}=1}^M q_\phi^{\tilde{m}}(\boldsymbol{z}_m|\boldsymbol{X})}{h(\boldsymbol{z}_m|\boldsymbol{X})} + \gamma \overset{!}{=} 0$$

$$\frac{\partial \mathcal{L}[h(\boldsymbol{z}_m|\boldsymbol{X}),\gamma]}{\partial \gamma} = \int_{\boldsymbol{z}_m} h(\boldsymbol{z}_m|\boldsymbol{X})d\boldsymbol{z}_m - 1 \overset{!}{=} 0$$

The first condition implies that for *any* value of $\boldsymbol{z}_m$, the ratio between the mixture distribution and the prior is constant, while the second condition demands that the prior be normalized. These conditions can only be met if the prior *equals* the mixture distribution, which proves the claim. $\qquad\square$

## 5 Experiments

We evaluate the proposed MMVM VAE on three benchmark datasets and two challenging real-world applications.

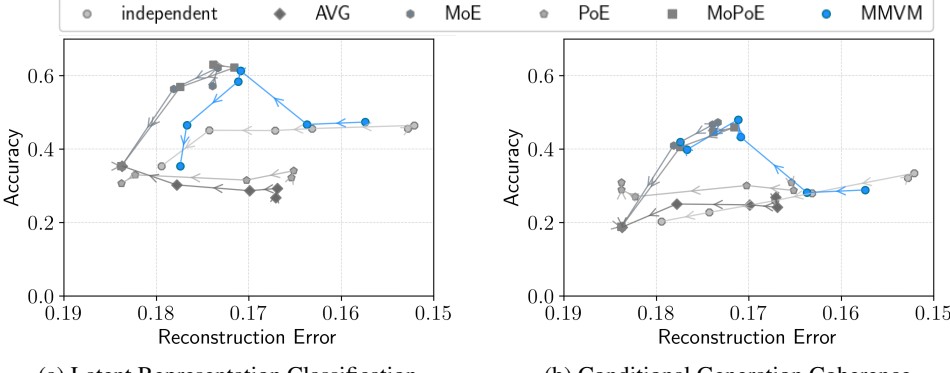

(a) Latent Representation Classification  (b) Conditional Generation Coherence

Figure 3: Results based on a memory experiment conducted on five rats, each regarded as a separate modality. We report the performance of the latent representation classification and the conditional generation coherence against the reconstruction loss for different $\beta$ values for the different VAE methods. Every point in the figures represents a specific $\beta$ value, where $\beta = (10^{-5}, 10^{-4}, 10^{-3}, 2.5 \times 10^{-3}, 5 \times 10^{-3}, 10^{-2})$. An optimal model would be in the top right corner.

## 5.1 Benchmark Datasets

We first compare the proposed method against five strong VAE-based learning approaches on three frequently used multimodal benchmark datasets[4].

**Datasets.** We perform benchmark experiments on the translated PolyMNIST [Daunhawer et al., 2022, Sutter et al., 2021], the bimodal CelebA [Sutter et al., 2020], and the CUB image-captions [Shi et al., 2019] dataset. The translated PolyMNIST dataset uses multiple instances of the MNIST dataset [LeCun et al., 1998] with different backgrounds but shared digits. The digits of the different modalities are randomly translated, so we cannot predict their location across modalities. Bimodal CelebA extends the CelebA dataset [Liu et al., 2015] with an additional text modality based on the attributes describing the faces. Similarly, the CUB image-captions dataset extends the Caltech bird dataset [Wah et al., 2011] with human-generated captions describing the images. Please see Appendix B for more details regarding the datasets.

**Baselines.** We evaluate our proposed method against a set of jointly-trained independent VAEs [*independent*, Kingma and Welling, 2014], different aggregation-based multimodal VAEs, and an aggregation-based multimodal VAE with additional modality-specific latent spaces. For the set of independent VAEs, there is no interaction or regularization between the different modalities during training. For the aggregation-based multimodal VAEs, we use a multimodal VAE with a joint posterior approximation function. We evaluate four different aggregation functions: a simple averaging [*AVG*, Hosoya, 2018], a product-of-experts [*PoE*, Wu and Goodman, 2018], a mixture-of-experts [*MoE*, Shi et al., 2019], and a mixture-of-products-of-experts [*MoPoE*, Sutter et al., 2021]. For the multimodal VAE with modality-specific subspaces, we use MMVAE+ method [*MMVAE+*, Palumbo et al., 2023]. We train all VAE methods as $\beta$-VAEs [Higgins et al., 2016], where $\beta$ is an additional hyperparameter weighting the rate term $R$ of the VAE (see Section 4). Appendix B.4 provides the implementation details of the proposed method and the baseline alternatives.

**Evaluation.** We test the methods' ability to infer meaningful representation when only a subset of modalities is available. In addition, we evaluate all methods in terms of their data imputation performance, where we withhold a subset of modalities at test time and conditionally generate them from the shared latent representations. In this imputation task, we assess whether the generated modalities are both of high quality and display the expected shared information, which we refer to as *coherence*. We assess the quality of the learned latent representations using linear classifiers trained on representations of the training set and the coherence using nonlinear classifiers trained on original samples of the training set[5]. We use the reconstruction error as a proxy for how well each method learns the underlying data distribution. We assess each method by relating their

---

[4]The code for the experiments on the benchmark datasets can be found here: `https://github.com/thomassutter/mmvmvae`.

[5]We use the same evaluation procedure as in previous work [e.g., Shi et al., 2019, Sutter et al., 2021]

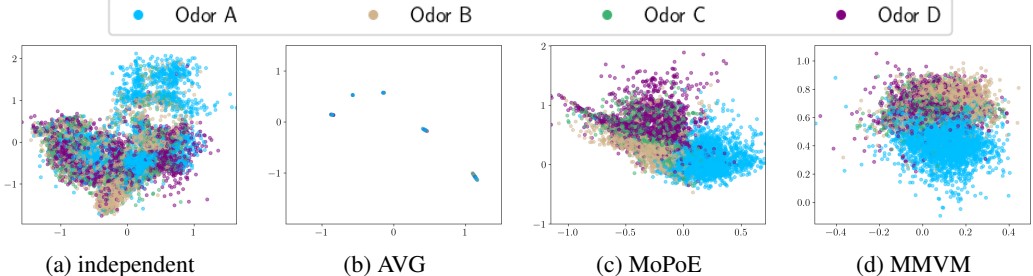

Figure 4: Latent neural representation during a memory experiment. Each model's performance is evaluated based on its own optimal $\beta$ value (0.00001, 0.01, 0.00001, 0.001 for independent, AVG, MoPoE, and MMVM respectively) in terms of the unimodal latent representation classification accuracy according to Figure 3a. Our method can distinguish the odor stimuli in the latent space with a clear separation of odors similar to MoPoE VAE (4 different colors). Conversely, unimodal and AVG models failed to combine multi-views as the odor separation only occurred within single views.

achieved reconstruction error to either the learned latent representation classification or the coherence (Figure 2). We evaluate all methods for multiple values of $\beta$, where the average performance over multiple seeds with a single $\beta$ value leads to a single point in Figure 2. Evaluating the methods for different values of $\beta$ considers that the optimal $\beta$ value is model- and data-dependent. In addition, increasing $\beta$ emphasizes a more structured latent space [Higgins et al., 2016]. Hence, highlighting the dynamics between reconstruction error and classification performance for different multimodal objectives provides additional insights. We chose $\beta \in \{2^{-8}, \ldots, 2^3\}$ on the PolyMNIST dataset, $\beta \in \{2^{-5}, \ldots, 2^4\}$ on the CelebA dataset, and $\beta \in \{2^{-2}, \ldots, 2^2\}$ on the CUB dataset. In all figures, the arrows go from small to large values of $\beta$. See Appendix B.2 for more details on the evaluation metrics.

**Results.** Figure 2 shows that the proposed MMVM VAE consistently outperforms the other VAE-based methods (*independent, AVG, MoE, PoE, MoPoE*) on all datasets and both tasks. We can show that our method overcomes the limitations of aggregation-based multimodal VAEs on translated PolyMNIST described in Daunhawer et al. [2020]. Also, MMVM VAE can learn meaningful representations and generate coherent samples across different modalities while achieving high reconstruction quality for both text-image datasets bimodal CelebA and CUB image-captions. The coherent conditional generation is especially surprising as the proposed MMVM VAE decoder is never confronted with a sample from another modality. For all benchmark datasets, the proposed MMVM either achieves better latent representation classification and coherence performance with a similar reconstruction loss or lower reconstruction loss with a similar classification performance than other multimodal VAE approaches. In summary, we can show that the newly proposed MMVM VAE overcomes the limitations of previous aggregation-based approaches to multimodal learning (see Section 2) and outperforms previous works on all three benchmark datasets. We provide more results on the benchmark datasets in Appendix B.

## 5.2 Hippocampal Neural Activities

**Dataset.** Temporal organization is crucial to memory, affecting various perceptual, cognitive, and motor processes. While we have made progress in understanding the brain's processing of the spatial context of memories, our knowledge of their temporal structure is still very limited. To this end, neuroscientists have recorded neural activity in the hippocampus of rats performing a complex sequence memory task [Allen et al., 2016, Shahbaba et al., 2022]. More specifically, this study investigates the temporal organization of memory and behavior by recording neural activity from the dorsal CA1 region of the hippocampus. Briefly, the task involves presenting rats with a repeated sequence of non-spatial events (four stimuli: odors A, B, C, D) at a single port [Shahbaba et al., 2022]. Since the same experimental setup was conducted across all rats, we consider the rats as different "modalities" and apply our proposed MMVM method to extract meaningful latent representations. While the existence (and importance) of subject-specific effects is well-known in neuroscience, such effects tend to be treated as unexplained variance because of the lack of the required analytical tools to extract and utilize this information properly.

Table 1: VAE latent representation quality evaluation. Average AUROC [in %] over three seeds of the two unimodal latent representations ($z_F$ and $z_L$) on a subset of MIMIC-CXR labels. The latent representations learned by the MMVM VAE lead to better classification performance compared to the other VAEs and are competitive with the fully-supervised method. Full results in Appendix B.7.3.

| | All labels | No Finding | Cardiomegaly | Edema | Lung Lesion | Pneumonia |
|---|---|---|---|---|---|---|
| *supervised* | *70.5* ±*12.1* | *73.0* ±*1.4* | ***80.3*** ±*1.4* | ***87.1*** ±*0.9* | *54.8* ±*2.5* | ***61.3*** ±*0.4* |
| independent | 68.0 ±8.3 | 75.3 ±1.4 | 73.5 ±2.8 | 79.2 ±3.9 | 60.1 ±1.2 | 55.8 ±0.8 |
| AVG | 69.8 ±8.5 | 76.3 ±1.5 | 76.1 ±2.4 | 81.3 ±3.3 | 60.4 ±1.4 | 57.3 ±0.6 |
| MoE | 68.9 ±8.7 | 76.5 ±0.7 | 74.9 ±1.6 | 80.2 ±2.3 | 59.6 ±1.3 | 56.9 ±1.0 |
| MoPoE | 70.3 ±8.9 | 77.2 ±0.2 | 76.3 ±0.8 | 82.1 ±1.2 | 60.8 ±0.6 | 57.8 ±0.7 |
| PoE | 70.4 ±8.3 | 75.9 ±1.3 | 76.7 ±1.9 | 81.8 ±2.7 | 61.3 ±2.1 | 57.8 ±0.4 |
| MMVM | **73.1** ±8.8 | **78.7** ±0.4 | 79.6 ±0.9 | 85.3 ±1.0 | **63.6** ±0.7 | 59.5 ±0.7 |

**Results.** Our proposed MMVM method outperforms[6] most previous works regarding learned latent representations and conditional generation coherence. Only the *MoPoE* VAE achieves a classification performance comparable to the MMVM method but with a higher reconstruction loss. Figure 3a shows the separation of the latent representation (measured by the accuracy of a multinomial logistic regression classifier) against the reconstruction loss. Similar to the results on the benchmark datasets, the proposed MMVM VAE outperforms previous works by providing a clear separation of odors in the latent space while maintaining a low reconstruction loss. Figure 3b compares the coherence of conditional generation accuracy against the reconstruction loss. As before, our proposed approach outperforms the alternatives. The proposed MMVM method allows learning an aligned latent representation across different modalities. We show the 2-dimensional latent representations for every rat and four VAE encoders in Figure 4. Each dot is the two-dimensional latent representation of a 100 ms sub-window of one odor trial for one rat and is colored according to its ground truth odor value. Figure 4 shows the odor stimuli separation on the latent space and how good MMVM VAE is in separating the odors. At the same time, two baseline models fail to extract the shared information between rats. Although it shows separation in some views, the independent model does not provide a connection between views. The five tiny clusters in Figure 4b show that, instead of showing a clear odor separation on the latent space, the *AVG* VAE separated the data by rats. In other words, the five rats' latent representations were far from each other, so the *AVG* VAE failed to connect the five views. See also Appendix B.8 for more results.

## 5.3 MIMIC-CXR

**Dataset.** To assess the performance of our approach in a real-world setting, we evaluate the proposed MMVM method on the automated analysis of chest X-rays, a common and critical medical task. For this purpose, we use the MIMIC-CXR dataset [Johnson et al., 2019b], a well-established and extensive collection of chest X-rays. The dataset reflects real clinical challenges with varying image quality due to technical issues, patient positioning, and obstructions. The dataset includes different views, which provide complementary information valuable for improving diagnostic [Raoof et al., 2012]. In this work, we consider frontal and lateral images as two modalities (see Appendix B.7 for further details). Each set of X-rays is labeled with different cardiopulmonary conditions, which have been automatically extracted from the associated radiology reports [Irvin et al., 2019]. This results in instances with incomplete label sets [Haque et al., 2023], which presents a challenge for fully supervised approaches and motivates the need for self-supervised methods instead.

**Results.** We evaluate[7] the quality of the unimodal latent representations of the MMVM VAE by comparing them with those learned by a set of jointly-trained independent VAEs (*independent*) as well as with representations from other multimodal VAEs that use aggregation-based approaches (*AVG*, *MoE*, *MoPoE*, and *PoE*) (see Section 5.1 for more details). We do this by training binary random forest classifiers independently for each method and all labels on the inferred representations of a

---

[6]The code for the hippocampal neural activity experiments can be found here: `https://github.com/yangmeng96/mmvmvae-hippocampal`

[7]The code for the MIMIC-CXR experiments can be found here: `https://github.com/agostini335/mmvmvae-mimic`

subset of the training set. Table 1 shows the AUROC for these classifiers, averaged over three seeds and both unimodal representations for a subset of labels. In addition, we also report the performance of a deep nonlinear network trained in a fully supervised manner (*supervised*) on the same train/test split for reference purposes. Detailed experiment information can be found in Appendix B.7.3, with extensive results for each modality and label available in Table 2 and Table 3. Overall, our approach shows performance improvements across all labels compared to the other VAEs and is highly competitive with the fully-supervised method, surpassing it in average performance over all labels. Examining each unimodal representation separately provides further insights into the VAEs' ability to leverage information from other modalities during training. For example, in the *Cardiomegaly* prediction task, the MMVM VAE's lateral representations $z_L$ slightly outperform the PoE VAE's frontal representations $z_F$ (MMVM $z_L$: 78.7%, PoE $z_F$: 78.5%), even though the lateral modality seems generally less informative (supervised $x_L$: 79.0%, $x_F$: 81.7%) for this task. The same observation can be made for other labels (see detailed results and discussion in Appendix B.7.3). This illustrates the MMVM VAE's ability to soft-share information between modality-specific latent representations during training, thereby enhancing the representation quality of the weaker modality.

## 6 Broader Impact & Limitations

This paper aims to advance the field of Machine Learning by providing a natural and fundamental solution for integrating data across modalities. The proposed approach can be applied to various scientific and engineering problems with a potentially significant societal impact. In the field of neuroscience specifically, our method could allow neuroscientists to leverage individual differences in brain activity and behavior to understand basic information processing in the brain, as well as to capture distinct longitudinal changes to understand how it is affected in disease. In translational research, it could help identify subjects more susceptible to disease or potentially more responsive to treatment. While we can show that the proposed method learns better representations and generates more coherent samples, we cannot directly generate random samples anymore (see also Section 4). Although we show results on two real-world datasets, Sections 5.2 and 5.3, additional experiments on even larger scale multimodal datasets would help further evaluate the proposed method, e.g. [Damen et al., 2018, Wu et al., 2023]. However, training and evaluating our methods on such datasets requires immense computing resources.

## 7 Discussion & Conclusion

In this work, we presented a new multimodal VAE, called MMVM VAE, based on a data-dependent multimodal variational mixture-of-experts prior. By focusing on a multimodal prior, the proposed MMVM VAE overcomes the limitations of previous methods with over-restrictive definitions of joint posterior approximations. The proposed objective leveraging the MMVM prior takes inspiration from contrastive learning objectives, where the goal is to minimize the similarity between positive pairs while maximizing the similarity between negative pairs [see, e.g., Chen et al., 2020, Tian et al., 2020]. In the MMVM objective, we minimize the similarity between different modalities of the same sample (positive pairs) via the regularizing term in the objective, whereas the second part of the objective, the reconstruction loss, prevents degenerate solutions.

In extensive experiments on three different benchmark datasets, we show that MMVM VAE outperforms previous works in terms of learned latent representations as well as generative quality and coherence of missing modalities. We also demonstrate its efficacy on two challenging real-world applications and show improved performance compared to previous VAEs and even a fully-supervised approach. Future research could involve studying the representation-distortion tradeoff from an information-theoretical perspective [Yang and Mandt, 2022, Yang et al., 2023] and applying similar ideas to more powerful multimodal generative models and representation learning methods. We see a lot of potential in applying the MMVM regularization to other multimodal and multiview objectives, e.g., as an additional guidance signal for diffusion models. While masked modeling has shown impressive results as an objective for representation learning, current multimodal masked modeling objectives concatenate the embedding tokens coming from different modalities [see, e.g., Bachmann et al., 2022]. Adding the MMVM regularization objective would offer an interesting alternative to sharing information from different modalities.

## Acknowledgements

TS, AA, and DC are supported by the grant #2021-911 of the Strategic Focal Area "Personalized Health and Related Technologies (PHRT)" of the ETH Domain (Swiss Federal Institutes of Technology). BS and NF received funding from the NIH award R01-MH115697 and NSF award NCS-FR-2319618. SM acknowledges support from the National Science Foundation (NSF) under an NSF CAREER Award IIS-2047418 and IIS-2007719, the NSF LEAP Center, by the Department of Energy under grant DE-SC0022331, the IARPA WRIVA program, the Hasso Plattner Research Center at UCI, and by gifts from Qualcomm and Disney.

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

# A  MMVM VAE

## A.1  Bound on the proposed objective

The objective $\mathcal{E}(\boldsymbol{x}_m)$ for a single modality $m$ is given by

$$\mathcal{E}(\boldsymbol{x}_m) = \mathbb{E}_{q_\phi^m(\boldsymbol{z}_m \mid \boldsymbol{x}_m)} \left[\log p_\theta(\boldsymbol{x}_m \mid \boldsymbol{z}_m)\right] - KL\left[q_\phi(\boldsymbol{z}_m \mid \boldsymbol{x}_m) \mid\mid p_\theta(\boldsymbol{z}_m)\right] \qquad (4)$$

$$\leq \mathbb{E}_{q_\phi^m(\boldsymbol{z}_m \mid \boldsymbol{x}_m)} \left[\log p_\theta(\boldsymbol{x}_m \mid \boldsymbol{z}_m)\right] \qquad (5)$$

$$\leq \log p_\theta(\boldsymbol{x}_m \mid \boldsymbol{\mu}_m(\boldsymbol{x}_m)), \qquad (6)$$

where $\boldsymbol{\mu}_m(\boldsymbol{x}_m) = f_m(\boldsymbol{x}_m)$ is the output of the optimized (mean) encoder $f_m(\cdot)$ of modality $m$. Equation (5) follows from the non-negativity of the KL divergence. Without regularization term, the maximizing distribution is a delta distribution with zero variance (Equation (6)). Equation (6) equals the maximum-likelihood version of the proposed MMVM VAE (for a single modality). Put differently, the MSE of a "vanilla" autoencoder is an upper bound on the objective $\mathcal{E}(\boldsymbol{x}_m)$ for any prior distribution $p_\theta(\boldsymbol{z}_m)$, hence, also for the newly introduced MMVM prior distribution $h(\boldsymbol{z}_m \mid \boldsymbol{X})$.

Let us have a look at when the KL term actually vanishes given a MMVM prior distribution $h(\boldsymbol{z}_m \mid \boldsymbol{X})$. The KL term can only vanish if all posterior approximation $q_\phi^{\tilde{m}}(\boldsymbol{z}_m \mid \boldsymbol{x}_{\tilde{m}})$ map to a single mode $q_\phi^m(\boldsymbol{z}_m \mid \boldsymbol{x}_m)$. In that case

$$h(\boldsymbol{z}_m \mid \boldsymbol{X}) = \frac{1}{M} \sum_{\tilde{m}=1}^{M} q_\phi^{\tilde{m}}(\boldsymbol{z}_m \mid \boldsymbol{x}_{\tilde{m}}) = q_\phi^m(\boldsymbol{z}_m \mid \boldsymbol{x}_m) \qquad (7)$$

and $KL\left[q_\phi^m(\boldsymbol{z}_m \mid \boldsymbol{x}_m) \mid\mid h(\boldsymbol{z}_m \mid \boldsymbol{X})\right] = 0$.

Another scenario is when $q_\phi^m(\boldsymbol{z}_m \mid \boldsymbol{x}_m)$ and $h(\boldsymbol{z}_m \mid \boldsymbol{x}_m)$ have disjoint modes. Hence, $q_\phi^m(\boldsymbol{z}_m \mid \boldsymbol{x}_m)$ is only a match to itself. In this case, we have

$$\mathbb{E}_{q_\phi^m(\boldsymbol{z}_m \mid \boldsymbol{x}_m)} \left[\log\left(\frac{1}{M} \sum_{\tilde{m}=1}^{M} q_\phi^{\tilde{m}}(\boldsymbol{z}_m \mid \boldsymbol{x}_{\tilde{m}})\right)\right] \approx \mathbb{E}_{q_\phi^m(\boldsymbol{z}_m \mid \boldsymbol{x}_m)} \left[\log\left(\frac{1}{M} q_\phi^m(\boldsymbol{z}_m \mid \boldsymbol{x}_m)\right)\right] \qquad (8)$$

$$= -\log M + \mathbb{E}_{q_\phi^m(\boldsymbol{z}_m \mid \boldsymbol{x}_m)} \left[\log q_\phi^m(\boldsymbol{z}_m \mid \boldsymbol{x}_m)\right] \quad (9)$$

So, our objective will still be (up to a constant) the maximum likelihood objective, leading the variances to shrink to zero. In this case, the objective will reduce to the limiting independent VAE, where the modalities do not "talk" to each other and there is no multimodal alignment.

In the most interesting case, there will be a non-trivial overlap between $q_\phi^m(\boldsymbol{z}_m \mid \boldsymbol{x}_m)$ and $h(\boldsymbol{z}_m \mid \boldsymbol{x}_m)$, i. e. between the different unimodal posterior approximations, leading to a multimodal alignment through the soft sharing that we wish to see.

In addition, Figure 5 empirically shows that the negative mean squared error (MSE) of the vanilla autoencoder (AE) upper bounds the proposed objective $\mathcal{E}$. We show that lowering the $\beta$ value of the regularizer $R$ (see Section 4) approximates the negative MSE of the vanilla AE.

# B  Experiments

## B.1  Dataset Licences

- PolyMNIST: originally published in [Sutter et al., 2021], downloaded the data from `https://drive.google.com/drive/folders/1lr-laYwjDq3AzalaIe9jN4shpt1wBsYM?usp=sharing` and the code from `https://github.com/thomassutter/MoPoE`, published under the MIT license

- Bimodal CelebA: originally published in [Sutter et al., 2020], downloaded from `https://drive.google.com/drive/folders/1lr-laYwjDq3AzalaIe9jN4shpt1wBsYM?usp=sharing`, licensed under the MIT license. The original CelebA dataset was published in [Liu et al., 2015], license not found.

- CUB image-captions: originally published in [Shi et al., 2019], downloaded from `http://www.robots.ox.ac.uk/~yshi/mmdgm/datasets/cub.zip`, licensed under GPL 3.0.

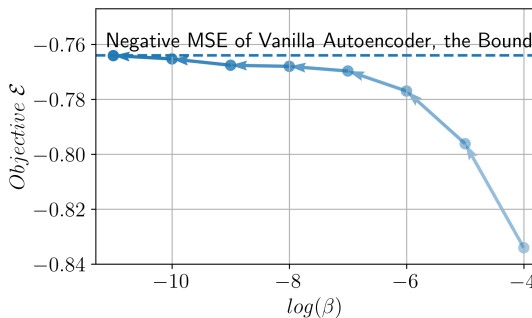

Figure 5: We compare the achieved values of the proposed objective $\mathcal{E}$ to the vanilla Autoencoder's negative mean squared error (MSE). Lowering the $\beta$ value of the regularizer $R$ in the objective (see Section 4) approximates the negative MSE bound provided by the vanilla AE. This proves empirically that the negative MSE of the vanilla AE indeed upper bounds the proposed objective $\mathcal{E}$.

- MIMIC-CXR: originally published in [Johnson et al., 2019b], downloaded from `https://physionet.org/content/mimic-cxr/2.0.0/`, licensed under PhysioNet Credentialed Health Data License 1.5.0 (see `https://physionet.org/content/mimic-cxr/view-license/2.0.0/`).

- MIMIC-CXR-JPG: originally published in [Johnson et al., 2019a], downloaded from `https://physionet.org/content/mimic-cxr-jpg/2.1.0/`, licensed under PhysioNet Credentialed Health Data License 1.5.0 (see `https://physionet.org/content/mimic-cxr-jpg/2.1.0/`).

- Hippocampal Neural Activity data: originally published in [Shahbaba et al., 2022], downloaded from `https://datadryad.org/stash/dataset/doi:10.7280/D14X30`, licensed under CC0 1.0 Universal (CC0 1.0) Public Domain Dedication (see `https://creativecommons.org/publicdomain/zero/1.0/`)

### B.2 Evaluation Details

We evaluate the different methods based on the coherence of their imputed samples, the quality of their latent representations, and their reconstruction error. We assume access to the full set of modalities during training, but we do not make this assumption at test time. Hence, there is a need for methods that can conditionally generate samples of these missing modalities, given the available modalities. In other words, we want to be able to impute missing modalities. Imputed modalities should not only be of high generative quality but also display the same shared information as the available modalities (i.e., be coherent). For example, if we generate a sample of modality $x_2$ based on modality $x_1$ from the PolyMNIST dataset (see Figure 6), the generated sample of modality $x_2$ should contain the same digit information as modality $x_1$ but show the background of modality $x_2$.

### B.2.1 Latent Representation Evaluation

We assess the learned representations based on subsets of modalities and not the full set. The quality of the representations serves as a proxy of how useful the learned representations are for additional downstream tasks that are not part of the training objective. Hence, high-quality representations of subsets of modalities are also the basis for conditionally generating coherent samples. We assess their quality by using the classification performance of linear classifiers. We train independent classifiers on the unimodal representations of the training set and evaluate them on unimodal test set representations. To assess the learned latent representation, we train a logistic regression classifier on 10000 latent representations of the training set. The accuracy is computed on all representations of the test set.

### B.2.2 Coherence Evaluation

The coherence of conditionally generated samples shows how well the content of the imputed modalities aligns with the content of the available modalities in terms of the shared information. We evaluate the coherence using ResNet-based classifiers [He et al., 2016] that are trained on samples

from the original training set of every modality. Using the described procedure, generated samples have to be visually similar to the original samples to have high coherence. Otherwise, the nonlinear classifier will not be able to predict digits correctly.

To compute the coherence of conditionally generated samples, we train additional deep classifiers on original samples of the training set. We use a ResNet-based non-linear classifier that is trained on the full original training set. The prediction of this classifier is then used to determine the class (on PolyMNIST: digit) of the conditionally generated sample of a missing modality. The nonlinear classifier reaches an accuracy of above 98% on the original test set. For other datasets, the nonlinear classifier is trained to predict the shared information of the multimodal dataset. Hence, it serves as a good oracle for determining the digit of generated samples.

**Conditional Generation with the MMVM VAE and the independent VAEs**   To generate modality $x_{\tilde{m}}$ conditioned on modality $x_m$, we proceed as follows:

1. We encode modality $x_m$ using the encoder $q_\phi^m(z_m \mid x_m)$
2. We sample a latent vector $z \sim q_\phi^m(z_m \mid x_m)$
3. We input the latent representation $z$ into the decoder $p_\theta^{\tilde{m}}(x_{\tilde{m}} \mid z)$ of modality $x_{\tilde{m}}$

To compute the coherence numbers reported, we perform the above steps for every modality $x_m$ where $m \in \{1, \ldots, M\}$ and average the achieved coherence accuracies.

**Conditional Generation with the aggregated multimodal VAEs**   All aggregation-based multimodal VAEs conditionally generate samples in the same way. Hence, the following procedure applies to the multimodal VAEs used in Section 5 ([AVG, Hosoya, 2018], [PoE, Wu and Goodman, 2018], [MoE, Shi et al., 2019], and [MoPoE, Sutter et al., 2021]). To generate modality $x_{\tilde{m}}$ conditioned on modality $x_m$, we proceed as follows:

1. We encode modality $x_m$ using the encoder $q_\phi^m(z_m \mid x_m)$
2. We sample a latent vector $z \sim q_\phi^m(z_m \mid x_m)$
3. We input the latent representation $z$ into the decoder $p_\theta^{\tilde{m}}(x_{\tilde{m}} \mid z)$ of modality $x_{\tilde{m}}$

If we would have access to a multimodal subset $X_A$ consisting of more than one modality, i.e. $|A| > 1$, we would have the joint posterior approximation distribution $q_\phi(z \mid X_A)$ of the subset $X_A$, sample a latent vector from that distribution, i.e. $z \sim q_\phi(z \mid X_A)$, and generate modality $x_{\tilde{m}}$ using the decoder $p_\theta^{\tilde{m}}(x_{\tilde{m}} \mid z)$ of modality $x_{\tilde{m}}$

### B.2.3   Generative Quality

The reconstruction error is a proxy for how well every method learns the underlying data distribution. We evaluate the different VAE models by their achieved reconstruction error against either the learned latent representation classification or coherence (e.g., Figure 2). We do this for multiple values of $\beta$, where the average performance over multiple seeds with a single $\beta$ value leads to a scatter point in the figures. This way, we can assess the trade-off between accurately reconstructing data samples and inferring shared information. In addition, we compute the FID [Heusel et al., 2017] values for modalities that can be summarized as natural images in the appendix. A low FID score correlates with a high-quality generated image.

### B.3   Implementation (General)

We use the scikit-learn [Pedregosa et al., 2011] package for the linear classifiers to evaluate the learned latent representations. All code is written using Python 3.11, PyTorch [Paszke et al., 2019] and Pytorch-Lightning [Falcon and The PyTorch Lightning team, 2019]. We base the implementations of the aggregation-based VAE methods on the official implementations. Hence, we base our implementations on the following repositories:

- AVG: `https://github.com/HaruoHosoya/gvae`
- PoE: `https://github.com/mhw32/multimodal-vae-public`

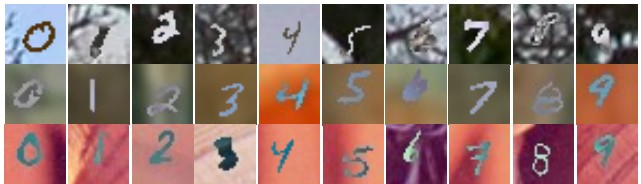

Figure 6: PolyMNIST (translated, scale=75%): every column is a multimodal tuple $\boldsymbol{X}$, and every row shows samples of a single modality $\boldsymbol{x}_m$. We see the random translation between samples by looking at images from a single row or column.

- MoE: `https://github.com/iffsid/mmvae`
- MoPoE: `https://github.com/thomassutter/MoPoE`

## B.4 PolyMNIST

### B.4.1 Dataset

The dataset is based on the original MNIST dataset [LeCun et al., 1998]. Compared to the original dataset, the digits are scaled down by a factor of $0.75$ such that there is more space for the random translation. In its original form, the PolyMNIST consists of 5 different modalities. We only use the first three modalities in this work. The background of every modality $\boldsymbol{x}_m$ consists of random patches of size $28 \times 28$ from a large image. The digit is placed at a random position of the patch. We refer to the original publication [Sutter et al., 2021] for details on the background images. Using this setup, every modality has modality-specific information given by its background image and shared information given by the digit, which is shared between all modalities. An additional difficulty compared to the original PolyMNIST is the random translation of the digits. The dataset can be found at `https://github.com/thomassutter/MoPoE`.

### B.4.2 Implementation & Training

We use the same network architectures for all multimodal VAEs. Every multimodal VAE consists of a ResNet-based encoder and a ResNet-based Decoder [He et al., 2016]. All modalities share the same architecture but are initialized differently. We assume Gaussian distribution for all unimodal posterior approximations, i.e.

$$q_\phi^m(\boldsymbol{z}_m \mid \boldsymbol{x}_m) = \mathcal{N}(\boldsymbol{z}_m; \boldsymbol{\mu}_m, \boldsymbol{\sigma}_m \boldsymbol{I}), \tag{10}$$

where the parameters $\boldsymbol{\mu}_m$ and $\boldsymbol{\sigma}_m$ are inferred using neural networks such that we have

$$q_\phi^m(\boldsymbol{z}_m \mid \boldsymbol{x}_m) = q_\phi^m(\boldsymbol{z}_m; \boldsymbol{\mu}_m(\boldsymbol{x}_m), \boldsymbol{\sigma}_m(\boldsymbol{x}_m)) = \mathcal{N}(\boldsymbol{z}_m; \boldsymbol{\mu}_m(\boldsymbol{x}_m), \boldsymbol{\sigma}_m(\boldsymbol{x}_m)) \tag{11}$$

The conditional data distributions $p_\theta(\boldsymbol{x}_m \mid \boldsymbol{z}_m)$ are modeled using the Laplace distribution, where the location parameter is modeled with a neural network (decoder) and the scale parameter is set to $0.75$ [Shi et al., 2019], i.e.

$$p_\theta(\boldsymbol{x}_m \mid \boldsymbol{z}_m) = \mathcal{L}(\boldsymbol{x}_m; \boldsymbol{\mu}_m, \boldsymbol{\sigma}_m), \tag{12}$$

where $\mathcal{L}(\cdot)$ defines a Laplace distribution. It follows that

$$p_\theta(\boldsymbol{x}_m \mid \boldsymbol{z}_m) = p_\theta(\boldsymbol{x}_m; \boldsymbol{\mu}_m(\boldsymbol{z}_m), \boldsymbol{\sigma}_m) = \mathcal{L}(\boldsymbol{x}_m; \boldsymbol{\mu}_m(\boldsymbol{z}_m), \boldsymbol{\sigma}_m) \tag{13}$$

We use the method of Hosoya [2018] for the implementation of the aggregated VAE. In this approach, a simplistic version of the joint posterior distribution is chosen where for Gaussian distribution joint posterior approximation $\mathcal{N}(\boldsymbol{\mu}_s, \boldsymbol{\sigma}_s \boldsymbol{I})$ we have the following distribution parameters $\boldsymbol{\mu}_s$ and $\boldsymbol{\sigma}_s$

$$\boldsymbol{\mu}_s = \frac{1}{M} \sum_{m=1}^{M} \boldsymbol{\mu}_m \tag{14}$$

$$\boldsymbol{\sigma}_s = \frac{1}{M} \sum_{m=1}^{M} \boldsymbol{\sigma}_s \tag{15}$$

$\boldsymbol{\mu}_m$ and $\boldsymbol{\sigma}_m$ are the distribution parameters of the unimodal posterior approximations $\mathcal{N}(\boldsymbol{\mu}_m, \boldsymbol{\sigma}_m \boldsymbol{I})$.

During training and evaluation, no weight-sharing takes place, i.e. every modality has its own encoder and decoder. We use the same architectures as in [Daunhawer et al., 2022]. For all experiments on this dataset, we use an Adam optimizer [Kingma and Ba, 2014] with an initial learning rate of 0.0005, and a batch size of 256. We train all models for 500 epochs.

We use NVIDIA GTX 2080 GPUs for all our runs. Each experiment can be run with 4 CPU workers and 16 GB of memory. An average run takes around 7 hours. To train all methods used in this paper, we had to train $11 \times 5 \times 6 = 330$ different models: 11 different $\beta$ values, 5 different seeds, and 6 different methods. Hence, the total GPU compute time used to generate the results for the PolyMNIST dataset is equal to $330 \times 7 = 2310$ hours. We—of course—also had to invest development GPU time, which we did not measure.

### B.4.3 Additional Results

We generated Figures 2a and 2d by plotting the classification accuracy of a linear classifier, which we trained on the learned latent representation, against the reconstruction error on the test set. For the learned latent representation, we train a classifier on the unimodal latent representations. For the aggregated VAE, this means that we train the classifier on samples of the unimodal posterior approximations and not the joint posterior approximations. Using this procedure, we test the different methods according to their performance in case of missing data, e.g., we only have access to a single modality instead of the full set at test time. For the reconstruction loss, however, we computed the error given the full set of modalities. The idea for Figures 2a and 2d is to compare the reconstruction quality (i.e., how well can we learn the data distribution?) against metrics that are related to the "generative factors" of the data and relate the different modalities to each other, i.e. the shared information of a multimodal dataset.

In Figure 7, we evaluate the performance of individual modalities in case of missing modalities. For that, we reconstruct every modality if it was the only modality available at test time. Hence, the modalities in the aggregated VAE have to be reconstructed based on the unimodal posterior approximations and not the joint posterior approximation. For the independent VAEs and the MMVM-VAE, the reconstruction of a modality is only based on its own unimodal posterior approximation. Hence, for the latter two methods, nothing changes in this setting. The performance of the learned latent representation and the coherence of generated samples are evaluated in the same way as in Figures 2a and 2d.

Figure 7 shows that the reconstruction error of the aggregated VAE increases a lot if every modality needs to reconstruct itself. Interestingly, we can see that the "self-reconstruction error" (the x-axis in Figures 7a and 7b) decreases with an increasing $\beta$-value, which is different to the other two methods and also different to the aggregated VAE's behavior in Figures 2a and 2d.

In addition, we also show the conditional FID values the different multimodal VAEs achieve. Conditional FIDs come from the conditional generation of modality $\boldsymbol{x}_m$ given $\boldsymbol{x}_{\bar{m}}$. From all the conditionally generated samples of modality $\boldsymbol{x}_m$, we then compute the conditional FID values. Figure 8 shows the aggregated values of all conditional FIDs. We again show the FID values in relation to the downstream task performance and the conditional generation coherence. We see that only the proposed MMVM VAE reaches FID values similar to the ones of a set of independent VAEs. However, the set of independent VAEs cannot achieve the same latent representation classification nor coherence as the proposed MMVM VAE.

### B.5 Bimodal CelebA

### B.5.1 Dataset

We take the dataset from Sutter et al. [2020]. It extends the original CelebA dataset [Liu et al., 2015] to the multimodal setting by creating a second modality based on the attributes. In the original dataset, every image comes with a set of 40 labels, such as blond hair, smiling, etc. The difficulty of the individual attributes is not only due to their visual appearance but also their frequency in the dataset. In the multimodal extension, Sutter et al. [2021] created a text string from the attribute words. Unlike the label vector, absent attributes are dismissed in the text string instead of negated. Additionally, the attributes are randomly ordered.

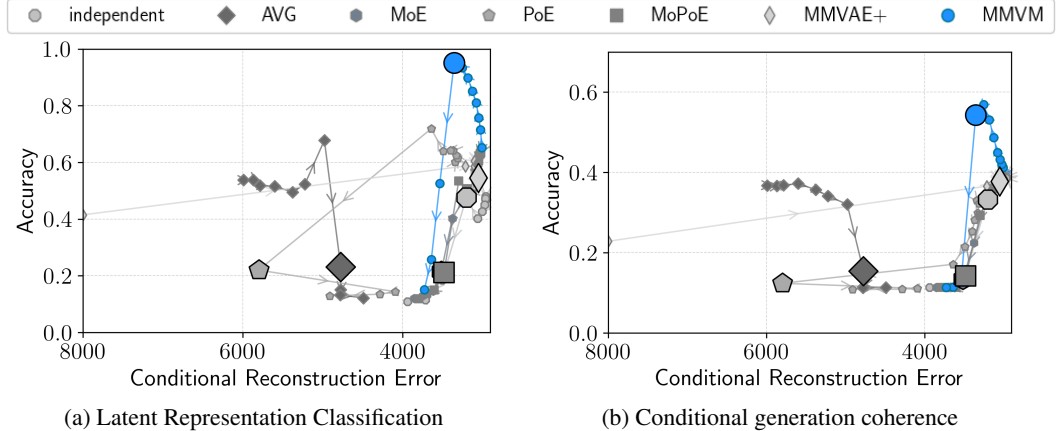

(a) Latent Representation Classification

(b) Conditional generation coherence

Figure 7: Results on the PolyMNIST dataset for different VAE methods. We report the performance of the latent representation classification and the conditional generation coherence against the conditional reconstruction loss for different $\beta$ values. Every point in the figures above is the average of five runs over different seeds and a specific $\beta$ value where $\beta = 2^k$ for $k \in \{-8, \dots, 3\}$. Different to Figures 2a and 2d, the x-axis is the sum of the self-reconstruction losses if only a single modality is given as input. Hence, for the aggregated VAE methods, every modality is decoded by its own unimodal posterior approximation instead of the joint posterior approximation.

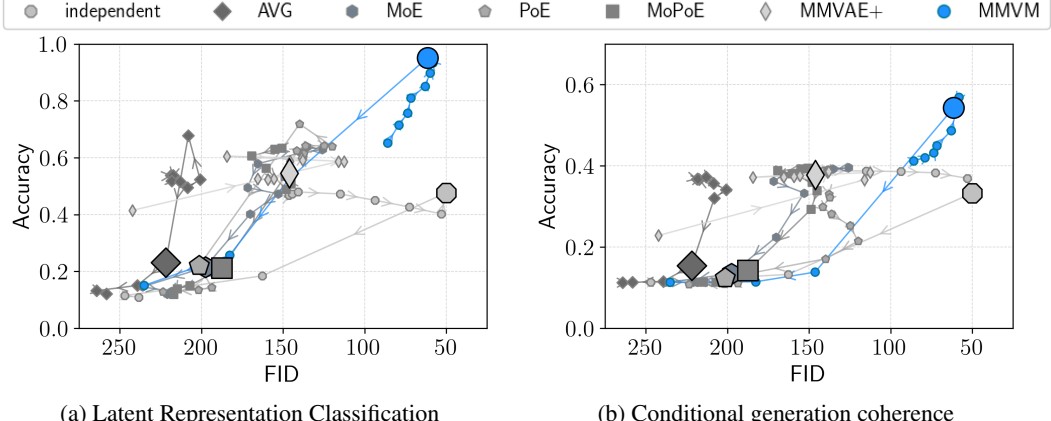

(a) Latent Representation Classification

(b) Conditional generation coherence

Figure 8: Results on the PolyMNIST dataset for different VAE methods. We report the performance of the latent representation classification and the conditional generation coherence against the conditional FID values. Every point in the figures above is the average of five runs over different seeds and a specific $\beta$ value where $\beta = 2^k$ for $k \in \{-8, \dots, 3\}$. Different to Figures 2a and 2d, the x-axis is not the reconstruction error but the average FID value computed from conditionally generated samples.

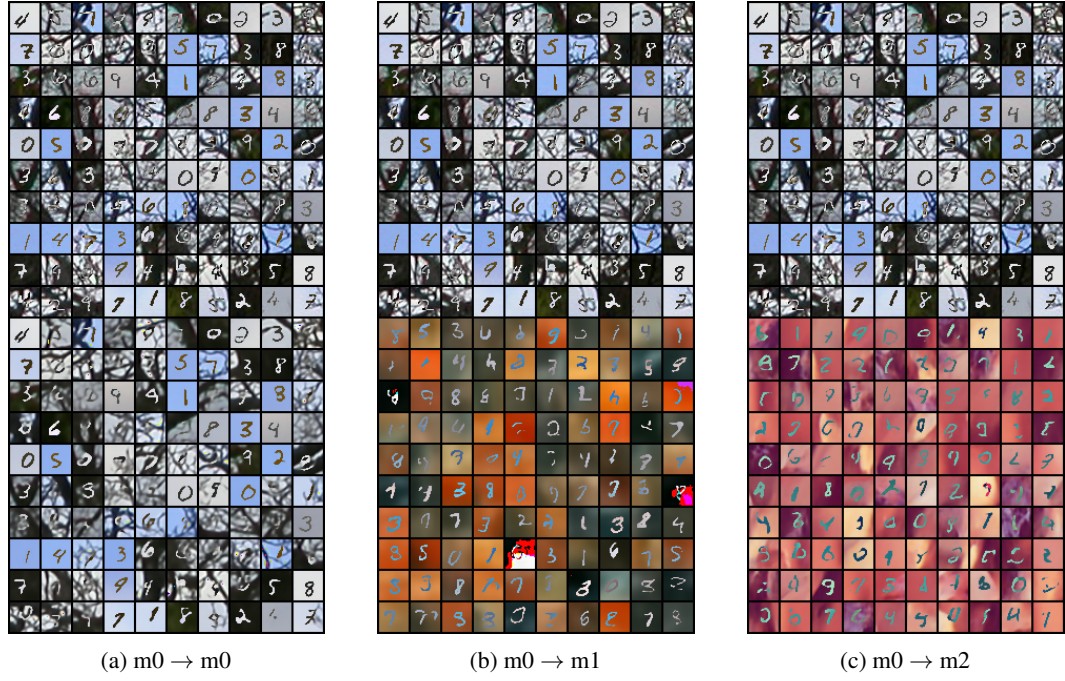

(a) m0 → m0          (b) m0 → m1          (c) m0 → m2

Figure 9: Qualitative results for the conditional generation task for the set of unimodal VAEs.

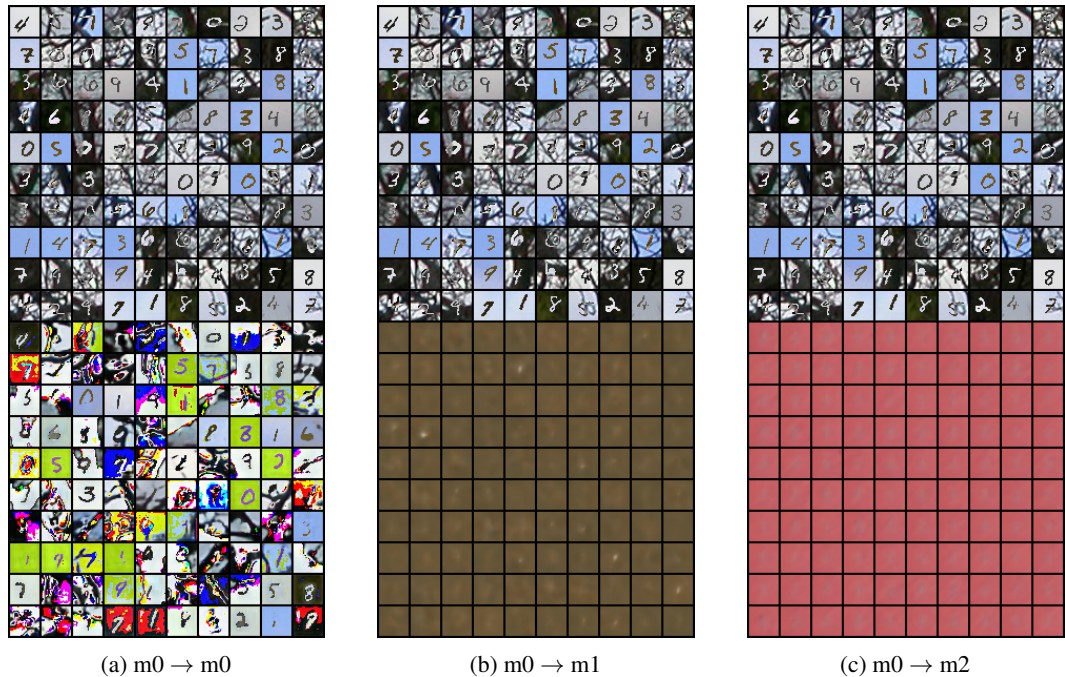

(a) m0 → m0          (b) m0 → m1          (c) m0 → m2

Figure 10: Qualitative results for the conditional generation task for the aggregation-based multimodal VAE (AVG).

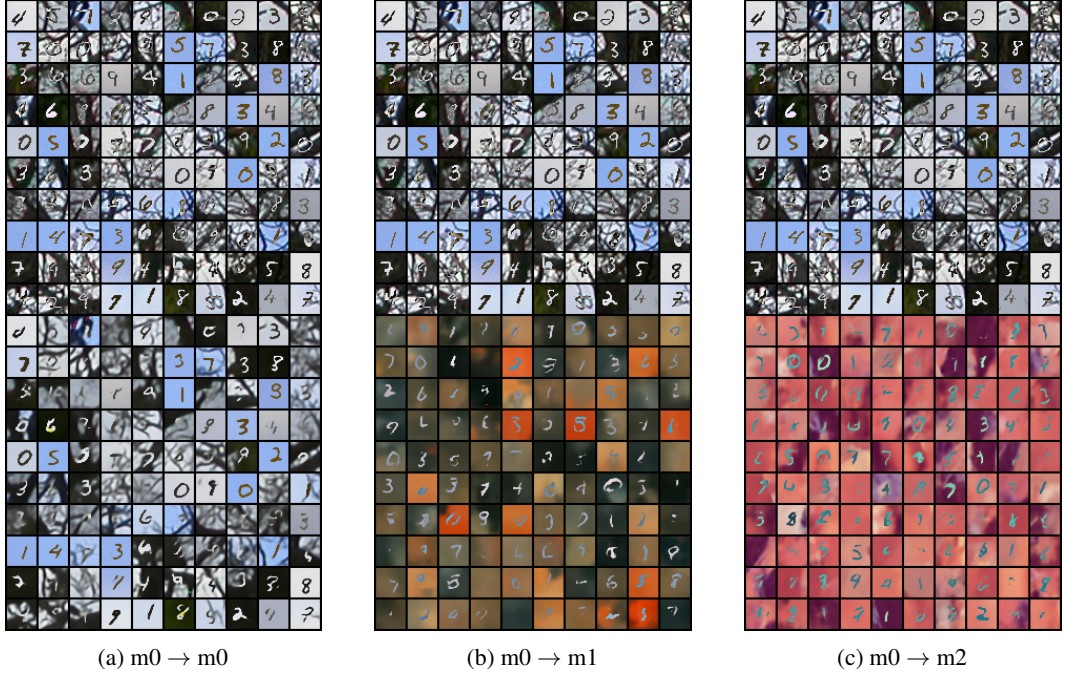

(a) m0 → m0          (b) m0 → m1          (c) m0 → m2

Figure 11: Qualitative results for the conditional generation task for the proposed MMVM VAE.

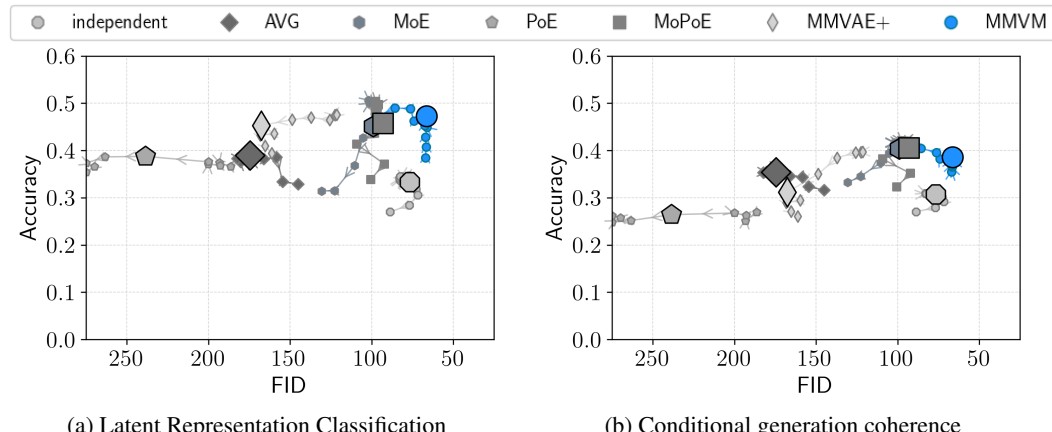

(a) Latent Representation Classification          (b) Conditional generation coherence

Figure 12: Results on the bimodal CelebA dataset for the different VAE methods. We report the performance of the latent representation classification and the conditional generation coherence against the conditional FID values. Every point in the figures above is the average of five runs over different seeds and a specific $\beta$ value where $\beta = 2^k$ for $k \in \{-5, \ldots, 4\}$. Unlike in Figures 2a and 2d, the x-axis is not the reconstruction error but the average FID value computed from conditionally generated samples.

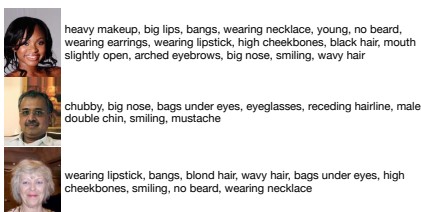

Figure 13: Bimodal CelebA. Three samples of image-text pairs. To introduce another level of difficulty to the text modality, we added a random translation to the starting point of the text attributes.

### B.5.2 Implementation & Training

We use ResNet-based encoders and decoders for this experiment as well [He et al., 2016], similar to the ones in the PolyMNIST experiment. The image encoder and decoder consist of 2D convolutions, while the text encoder and decoder consist of 1D convolutions. We use a character-level encoding and not a word or token-level encoding because of the synthetic nature of the text modality. We also use an Adam optimizer [Kingma and Ba, 2014] with a starting learning rate of 0.0002 and a batch size of 128. We train all models for 400 epochs and 3 seeds. The implementation follows the one described in Appendix B.4.

We use NVIDIA GTX 2080 GPUs for all our runs. Each experiment can be run with 4 CPU workers and 16 GB of memory. An average run takes around 24 hours. To train all methods used in this paper, we had to train $10 \times 3 \times 6 = 180$ different models: 10 different $\beta$ values, 3 different seeds, and 6 different methods. Hence, the total GPU compute time used to generate the results for the PolyMNIST dataset equals $180 \times 24 \approx 4320$ hours. We—of course—also had to invest GPU time to develop the method, which we did not measure.

### B.5.3 Additional Results

Given the multilabel nature of the CelebA dataset, we evaluate the learned latent representation with respect to the individual attributes and not only the average performance across all attributes. Figure 14 shows the detailed results according to the full set of 40 attributes for the three methods: independent VAEs, aggregated VAE, and MMVM VAE. We again train linear binary classifiers on inferred representations of the training set and evaluate them on representations of the test set. However, we now report the individual performance of every classifier. In the main text (see Figures 2b and 2e), we report the average performance of the 40 binary classifiers.

### B.6 CUB Image-Captions

### B.6.1 Dataset

The dataset contains 11,788 images of birds and 117,880 (10 times as many) captions, each image with 10 fine-grained captions describing the bird's appearance characteristics collected through Amazon Mechanical Turk (AMT). We performed a 75-25 training-test split, with 8,855 and 2,933 images, and 88,550 and 29,330 corresponding captions in the training set and test set, respectively. We resized the images to 3 channels and 64 by 64 pixels and modeled the captions using embeddings.

The labels applied to downstream tasks are based on the bird's primary colors. Based on the primary color attribute, we remapped the original 15-class primary color label set into a 6-class label set to address the sparsity issue. The labels and mappings are blue-to-red (blue, iridescent, purple, green, pink, red), brown (brown), grey (grey), yellow (yellow, olive, orange, buff), black (black), and white (white).

### B.6.2 Implementation Details & Training

The network structures are similar to the ones in other experiments. For both image and caption data, we employed a CNN encoder and decoder. We used a 128-dimensional latent space with a Laplace likelihood on images and a Categorical likelihood for captions. The activation function was chosen to be ReLU between layers. For all experiments on this dataset, we used an Adam optimizer with an initial learning rate of 0.001 and a batch size of 128. We trained all models for 100 epochs.

### B.7 MIMIC-CXR

### B.7.1 Dataset

The dataset we use in our experiment is a multimodal interpretation of the original MIMIC-CXR dataset. The original MIMIC-CXR dataset comprises high-resolution chest X-ray images related to imaging studies. A study may include multiple chest X-ray images captured from several view positions. We categorized these views into two primary modalities: *frontal* (including "AP" and "PA" views) and *lateral* (including "LL" and "Lateral" views). For each study, we pair every frontal image with every lateral image in all possible combinations. Studies lacking at least one frontal and one

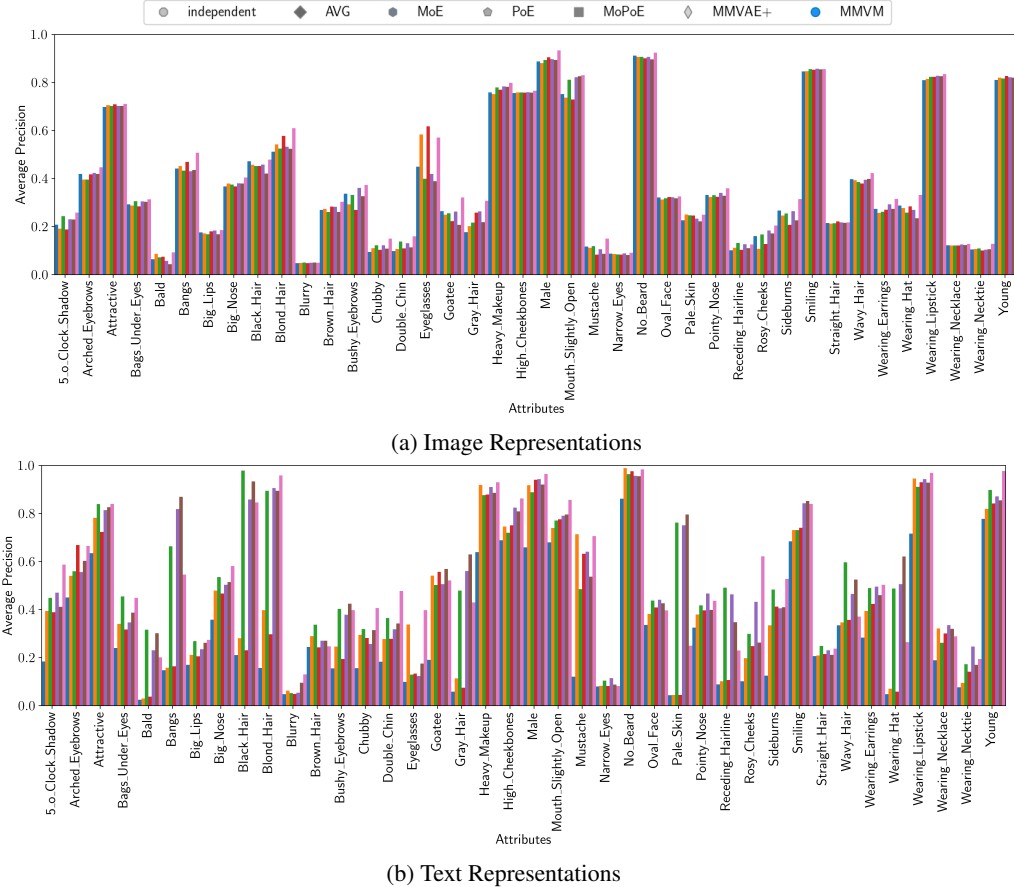

(a) Image Representations

(b) Text Representations

Figure 14: Attribute-level results on the bimodal CelebA datasets for the latent representation classification. The MMVM VAE outperforms the independent VAEs and the aggregated VAE on most attributes.

lateral image are excluded. This approach formalizes a new dataset composed of image pairs, thus offering a bimodal interpretation of the original MIMIC-CXR dataset. More rigorously, we define a dataset $\mathbb{X} = \{\boldsymbol{X}^{(i)}\}_{i=1}^n$ where each $\boldsymbol{X}^{(i)} = \{\boldsymbol{x}_f^{(i)}, \boldsymbol{x}_l^{(i)}\}$ is a bimodal tuple composed of one frontal image and one lateral image of the same study. An image may appear in multiple tuples, but we ensure that each tuple is unique by having at least one different image. Examples of these bimodal tuples are illustrated in Figure 17.

We use the JPG version of the MIMIC-CXR dataset, namely the MIMIC-CXR-JPG dataset [Johnson et al., 2019a]. In our preprocessing pipeline, we apply center cropping and downscale the images to a resolution of $224 \times 224$. We utilize the labels from the MIMIC-CXR-JPG dataset which are obtained using the CheXpert tool [Irvin et al., 2019]. All non-positive labels (including "negative," "non-mentioned," or "uncertain") were combined into an aggregate "negative" label following the approach adopted by Haque et al. [2023]. Each imaging study is connected to a subject. We split the dataset into distinct training (80%), validation (10%), and test (10%) sets based on subjects, thus ensuring that the same image or study cannot be present in multiple sets.

### B.7.2 Implementation & Training

**Multimodal VAEs Implementation & Training**    We use ResNet-based encoders and decoders for this experiment [He et al., 2016], similar to those used in the PolyMNIST and the Bimodal CelebA experiments. The image encoder and decoder consist of 2D convolution layers. The architectural design of the image encoders and decoders is uniform for both frontal and lateral modalities. We use an Adam optimizer [Kingma and Ba, 2014] with a learning rate of 0.00005 and a batch size of 32.

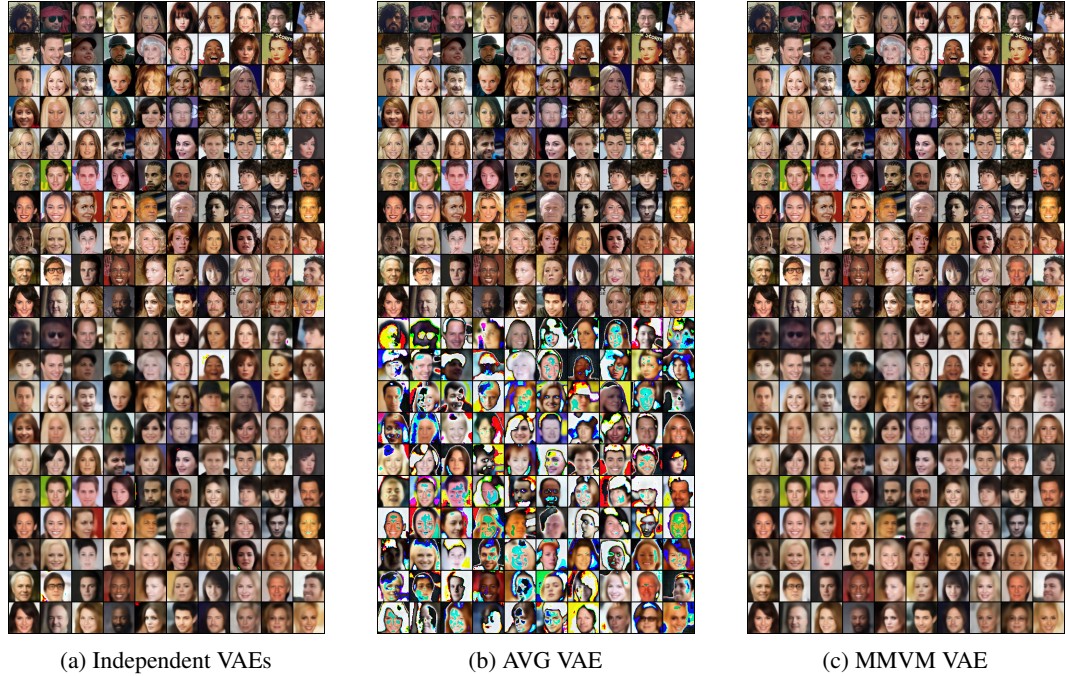

| (a) Independent VAEs | (b) AVG VAE | (c) MMVM VAE |

Figure 15: Qualitative Results for the CelebA dataset on the image-to-image generation task. The first 10 rows of every subplot show the input image and the bottom 10 rows its conditional generation. Different to the training, we provide only the image to every model and based on the latent representation of that image, we generate a sample. We see that the aggregated VAE (Figure 15b) is not able to conditionally generate visually pleasing samples compared to the independent VAEs (Figure 15a) and the MMVM VAE (Figure 15c).

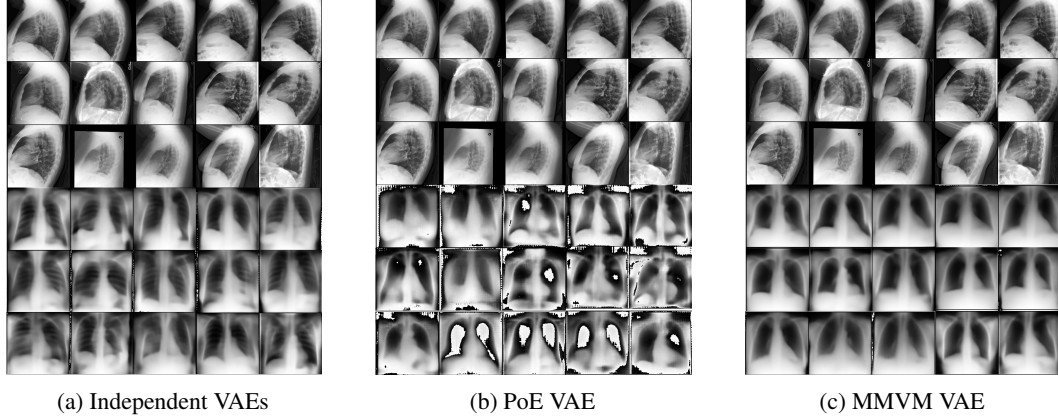

| (a) Independent VAEs | (b) PoE VAE | (c) MMVM VAE |

Figure 16: Qualitative results for the MIMIC-CXR dataset on the conditional generation task lateral to frontal. Results are aligned with the other datasets.

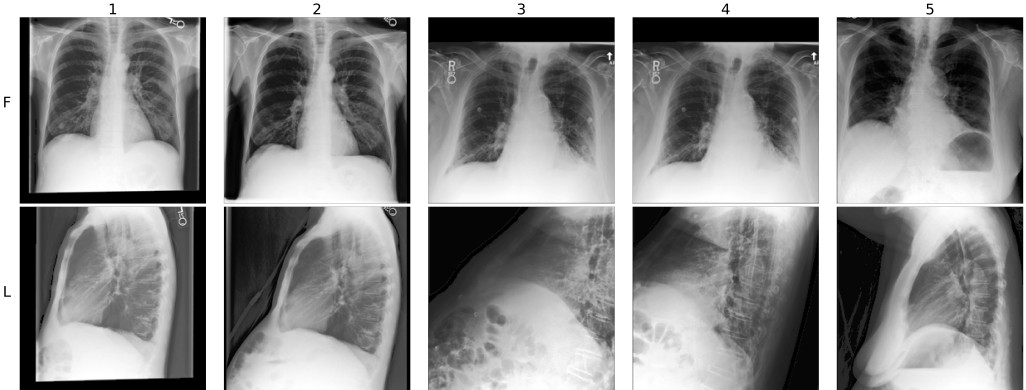

Figure 17: MIMIC-CXR experiment dataset: every column is a bimodal tuple $\boldsymbol{X}$, the top row shows samples of the frontal modality $\boldsymbol{x}_f$, and the bottom row shows samples of the lateral modality $\boldsymbol{x}_l$. The first two tuples are linked to *No Findings*, indicative of healthy conditions. Tuples three and four are labeled with *Consolidation* disease. The tuple five is labeled with *Atelectasis* disease. We can observe that tuples three and four share the same frontal image, but they differ due to having distinct lateral images.

We train all methods for 240 epochs and 3 seeds. The implementation follows the one described in Appendix B.4.

We use NVIDIA A100-SXM4-40GB GPUs for all our runs. An average run, evaluating one method on one seed, takes approximately 45 hours. To train all methods evaluated in this experiment, we had to train $3 \times 3 = 9$ different models: 3 seeds and 3 methods. Hence, the total GPU compute time used to generate the VAE results for the MIMIC-CXR experiment is around $45 \times 9 = 405$ hours. We also had to use GPU time in the development process, which we did not measure.

**Supervised classifier Implementation & Training**   We use supervised classifiers based on Resnet. The classifier architecture is derived from the VAEs encoder we use in this experiment. To adapt the encoder for classification tasks, we added a linear layer equipped with 14 neurons corresponding to the number of labels in the MIMIC-CXR dataset. We use an Adam optimizer with a learning rate of 0.0001 and a batch size of 256. We train distinct models for both frontal and lateral modalities, training each for 30 epochs and 3 seeds.

We train the classifiers using NVIDIA A100-SXM4-40GB GPUs. To train one classifier on one seed takes approximately 1 hour. We train a total of $3 \times 2 = 6$ classifiers: 3 seeds and 2 modalities, resulting in about 6 hours of GPU compute time. We also had to use GPU time in the development process, which we did not measure.

### B.7.3   Additional Results

In our main experiment, we evaluate the quality of the learned latent representations using Random Forest (RF) classifiers. Specifically, we train independent RF binary classifiers for each model and each label on the inferred representations of the training set and evaluate them on the representations of the test set. The RF classifiers are configured with 5,000 estimators and a maximum depth of 30. In Table 1 in Section 5.3, we report for each model the average performance over the two modalities and three seeds, totaling six RF classifiers per model: one for each modality and each of the three different seeds.

Here, we provide detailed insights into the capabilities of the models to leverage different modalities during training to improve the unimodal representations by reporting the performance of the latent representation classification for each modality separately in Table 2 and Table 3. These results allow us to highlight the strengths of the MMVM model in improving unimodal representations with additional modalities. For instance, in the main text we mentioned that in the Cardiomegaly classification task, the MMVM lateral representations $z_L$ lead to substantial improvements over the frontal representations $z_F$ of the other VAEs, even though the former modality $x_L$ is less informative than the latter $x_F$ as indicated by the respective performance of the supervised model on each

Table 2: Evaluation of the VAEs' frontal latent representations $z_F$ classification performance on the test split. The performance of a fully-supervised non-linear deep network is included for reference. The average AUROC [%] and standard deviation over three seeds are reported. Enl. Cardiom. stands for Enlarged Cardiomediastinum and Support Dev. for Support Device.

| | supervised | independent | AVG | MoE | MoPoE | PoE | MMVM |
|---|---|---|---|---|---|---|---|
| Atelectasis | **79.5** ±0.3 | 73.1 ±0.0 | 75.2 ±0.3 | 73.0 ±0.5 | 74.2 ±0.4 | 75.7 ±0.3 | 77.6 ±0.1 |
| Cardiomegaly | **81.7** ±0.1 | 76.3 ±0.4 | 78.5 ±0.2 | 76.5 ±0.6 | 77.1 ±0.1 | 78.5 ±0.3 | 80.5 ±0.1 |
| Consolidation | 65.3 ±0.7 | 62.4 ±0.4 | 66.0 ±0.8 | 62.9 ±0.6 | 63.9 ±0.3 | 66.7 ±0.8 | **69.1** ±0.6 |
| Edema | **88.0** ±0.2 | 83.0 ±0.3 | 84.6 ±0.3 | 82.4 ±0.6 | 83.1 ±0.6 | 84.5 ±0.3 | 86.3 ±0.1 |
| Enl. Cardiom. | 57.9 ±1.3 | 59.5 ±1.2 | 64.9 ±0.8 | 61.7 ±0.5 | 64.1 ±0.5 | 66.4 ±0.5 | **68.6** ±1.1 |
| Fracture | 51.4 ±0.5 | 56.0 ±0.1 | 58.4 ±0.5 | 57.3 ±0.2 | 57.4 ±0.7 | **58.8** ±0.3 | 58.6 ±0.8 |
| Lung Lesion | 52.4 ±0.4 | 61.3 ±0.4 | 61.8 ±0.2 | 60.6 ±0.9 | 60.7 ±0.8 | 63.4 ±0.4 | **64.1** ±0.2 |
| Lung Opacity | **69.5** ±0.2 | 63.8 ±0.3 | 65.7 ±0.3 | 63.5 ±0.2 | 64.7 ±0.0 | 66.3 ±0.2 | 68.1 ±0.1 |
| No Finding | 73.9 ±1.3 | 76.6 ±0.3 | 77.8 ±0.0 | 77.1 ±0.2 | 77.4 ±0.1 | 77.2 ±0.2 | **79.1** ±0.1 |
| Pleural Effusion | **88.0** ±0.0 | 81.2 ±0.6 | 82.8 ±0.0 | 81.6 ±0.4 | 82.5 ±0.5 | 82.8 ±0.2 | 85.7 ±0.3 |
| Pleural Other | 53.9 ±1.0 | 67.8 ±1.1 | 68.9 ±0.5 | 67.5 ±1.0 | 68.3 ±1.2 | 68.7 ±1.2 | **70.0** ±2.0 |
| Pneumonia | **61.3** ±0.3 | 55.3 ±0.5 | 57.8 ±0.4 | 56.4 ±0.4 | 57.3 ±0.4 | 57.5 ±0.4 | 60.0 ±0.6 |
| Pneumothorax | 76.9 ±1.1 | 75.3 ±1.0 | 78.3 ±0.6 | 77.7 ±0.4 | 78.8 ±0.7 | 78.5 ±0.7 | **81.9** ±0.4 |
| Support Dev. | **77.7** ±0.1 | 70.8 ±0.3 | 73.1 ±0.4 | 72.7 ±0.7 | 73.7 ±0.6 | 73.8 ±0.1 | 76.6 ±0.2 |
| All Labels | 69.8 ±12.6 | 68.7 ±9.0 | 71.0 ±8.6 | 69.4 ±8.8 | 70.2 ±8.8 | 71.3 ±8.4 | **73.3** ±8.9 |

Table 3: Evaluation of the VAEs' lateral latent representation $z_L$ classification performance on the test split. The performance of a fully-supervised non-linear deep network is included for reference. The average AUROC [%] and standard deviation over three seeds are reported. Enl. Cardiom. stands for Enlarged Cardiomediastinum and Support Dev. for Support Device.

| | supervised | independent | AVG | MoE | MoPoE | PoE | MMVM |
|---|---|---|---|---|---|---|---|
| Atelectasis | **78.0** ±0.1 | 70.7 ±0.3 | 73.5 ±0.4 | 72.8 ±0.1 | 74.7 ±0.2 | 73.7 ±0.2 | 77.0 ±0.2 |
| Cardiomegaly | **79.0** ±0.2 | 70.8 ±0.9 | 73.7 ±0.1 | 73.3 ±0.2 | 75.5 ±0.1 | 74.8 ±0.1 | 78.7 ±0.0 |
| Consolidation | 68.6 ±1.4 | 64.4 ±1.4 | 65.4 ±1.5 | 64.9 ±0.9 | 65.8 ±0.8 | 66.7 ±0.9 | **70.2** ±0.8 |
| Edema | **86.2** ±0.1 | 75.4 ±0.9 | 78.0 ±0.3 | 78.0 ±0.5 | 81.1 ±0.8 | 79.1 ±0.1 | 84.3 ±0.3 |
| Enl. Cardiom. | 61.5 ±1.0 | 60.1 ±0.7 | 62.0 ±1.0 | 60.5 ±0.5 | 64.2 ±0.9 | 63.5 ±0.8 | **69.0** ±0.7 |
| Fracture | 52.3 ±0.2 | 57.9 ±0.6 | 58.3 ±0.7 | 56.8 ±0.8 | 58.6 ±0.8 | 59.0 ±0.5 | **60.9** ±0.3 |
| Lung Lesion | 57.3 ±0.4 | 58.9 ±0.2 | 59.0 ±0.2 | 58.6 ±0.8 | 60.8 ±0.3 | 59.3 ±0.3 | **63.0** ±0.7 |
| Lung Opacity | **68.9** ±0.2 | 61.9 ±0.5 | 63.4 ±0.4 | 63.9 ±0.1 | 65.4 ±0.4 | 64.1 ±0.4 | 68.1 ±0.2 |
| No Finding | 72.0 ±0.8 | 73.9 ±0.3 | 74.8 ±0.2 | 75.9 ±0.2 | 77.1 ±0.1 | 74.6 ±0.1 | **78.3** ±0.1 |
| Pleural Effusion | **91.0** ±0.3 | 80.2 ±0.2 | 82.0 ±0.1 | 82.0 ±0.3 | 84.3 ±0.2 | 82.1 ±0.1 | 85.7 ±0.1 |
| Pleural Other | 61.5 ±1.9 | 62.8 ±1.3 | 64.3 ±0.7 | 62.7 ±1.7 | 63.6 ±1.0 | 63.9 ±1.0 | **68.5** ±1.9 |
| Pneumonia | **61.2** ±0.4 | 56.4 ±0.5 | 56.9 ±0.3 | 57.5 ±1.2 | 58.3 ±0.5 | 58.2 ±0.1 | 59.0 ±0.2 |
| Pneumothorax | **82.4** ±0.8 | 75.6 ±0.5 | 77.8 ±0.3 | 76.9 ±0.7 | 79.2 ±0.6 | 78.6 ±0.2 | 81.7 ±0.3 |
| Support Dev. | **77.3** ±0.1 | 71.9 ±0.6 | 72.9 ±0.5 | 73.6 ±0.7 | 75.9 ±0.4 | 74.7 ±0.5 | 77.1 ±0.3 |
| All Labels | 71.2 ±11.3 | 67.2 ±7.6 | 68.7 ±8.1 | 68.4 ±8.4 | 70.3 ±8.6 | 69.4 ±8.0 | **73.0** ±8.5 |

modality. This behavior is not specific to Cardiomegaly and can be observed for other labels such as Atelectasis, Lung Opacity, No Finding, Pleural Effusion, Pneumonia, and Support Devices. Even though the difference between modalities for these labels is sometimes small, the consistency of those results demonstrate the soft-sharing capabilities of the MMVM approach. Conversely, in tasks where the frontal view is less informative, such as Consolidation, Enlarged Cardiomediastinum, and Pneumothorax, the MMVM's lateral representations outperform the frontal ones of the other VAEs.

The efficacy of the soft sharing mechanism in the MMVM VAE is also reflected in the average performance for all labels. The performance difference between the two unimodal latent representations produced by the MMVM VAE is substantially smaller (0.9 percentage points on average) compared to the independent VAEs (2.5 percentage points on average) and the PoE VAE (2.2 percentage points on average).

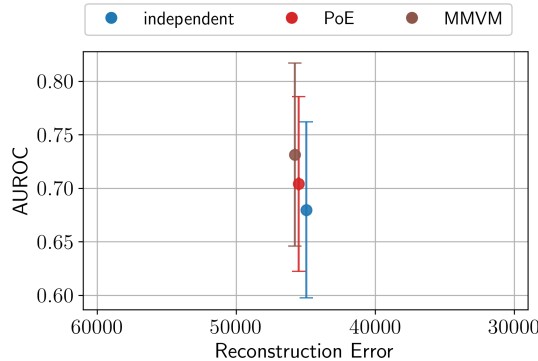

Figure 18: Latent representation classification for the MIMIC-CXR dataset. The mean AUROC over all labels and averaged over three seeds is reported.

Figure 18 illustrates the performance of the average unimodal latent representation $z$ classification described above against the reconstruction loss for the three different VAE methods introduced in the main text (independent, aggregate PoE VAE and the proposed MMVM VAE). As a reminder, the AUROC performance [in %] is averaged over three seeds and the two modalities.

## B.8 Hippocampal Neural Activities

### B.8.1 Dataset

The training data was collected from 250 ms length time frames after the port entry. Due to the behavior difference from each rat (some rats react faster to the odors while some others react slower), the training time frames of the five rats started from 250 ms, 250 ms, 500 ms, 500 ms, and 250 ms, respectively. During training, we treated each data point as independent and trained all the VAE models based on sliding windows (100 ms sub-window, 10 ms steps; 16 data points per window on each trial). The 100 ms sub-windows constituted the input data, with the dimension equal to the rat's number of neurons multiplied by 10, as the data was further binned into 10-ms increments.

### B.8.2 Implementation & Training

We use the same network architectures for all multimodal VAEs. Each of the autoencoders includes its unique encoder and decoder, both containing two hidden layers, without weight-sharing during training and evaluation. All modalities share the same architecture but the layers' dimensions are different, with 920, 790, 1040, 490, 460 dimensional input and hidden layers, respectively. The activation function was chosen to be LeakyReLU with a 0.01 negative slope. For all experiments on this dataset, we use an Adam optimizer with an initial learning rate of $0.001$, a batch size of $128$. We train all models for $1000$ epochs.

### B.8.3 Additional Results

We show the 2-dimensional latent representations for every rat and the six VAE encoders in Figure 4 and Figure 20. In these two figures, each dot is the two-dimensional latent representation of a 100 ms sub-window of one odor trial for one rat. Figure 4 is colored by 4 odors, and Figure 20 is colored by 5 modalities (rats). Figure 4 shows the odor stimuli separation on the latent space and how good MMVM VAE is in separating the odors. Figure 20 shows that the proposed MMVM VAE can best utilize the shared information between the five rats by pulling the latent representations together. At the same time, the independent AVG and PoE baseline models fail to extract the information shared between rats. Although it shows separation in some views, the independent model does not provide a connection between views. The five tiny clusters in Figure 4 and Figure 20 show that, instead of showing a clear odor separation on the latent space, the AVG model separated the data by rats. The results went against the intention to share information across views. In other words, the five rats' latent representations were far away from each other, so the aggregated VAE completely failed to connect the five views.

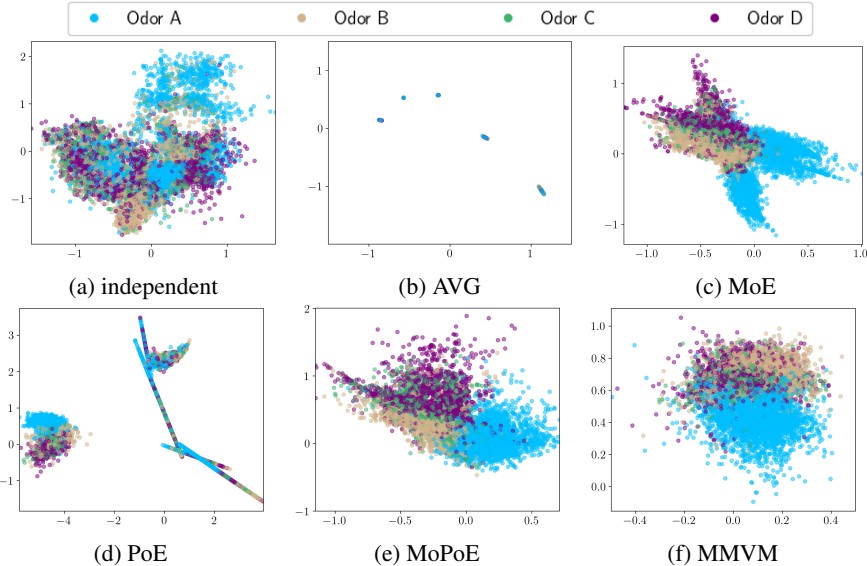

Figure 19: Latent neural representation during a memory experiment. Each model's performance is evaluated based on its own optimal $\beta$ value (0.00001, 0.01, 0.001 for independent, aggregated, and MMVM respectively) in terms of the self-conditioned latent representation classification accuracy according to Figure 3a. Our model can distinguish the odor stimuli in the latent space with a clear separation of odors (4 different colors).

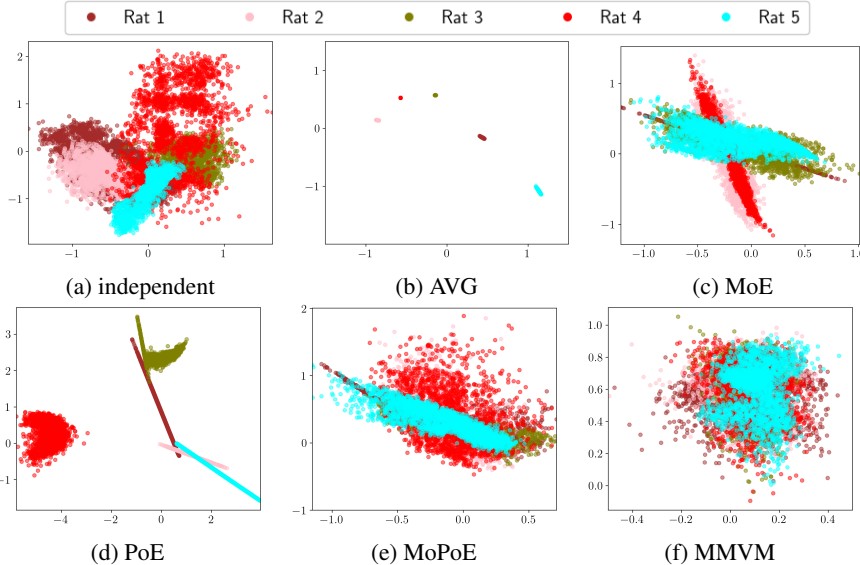

Figure 20: Latent Representation of Rats Brain by Each Rat. In our proposed MMVM model, the five views shared latent representations as the latent representation of all five views (colors) gathered together, while the two baseline models failed to combine multi-views.

