# OpenReview forum: "Unity by Diversity: Improved Representation Learning for Multimodal VAEs"
_NeurIPS.cc/2024/Conference — NeurIPS 2024 poster_

### Official Review · Reviewer_6TPP · 2024-07-01

**Soundness:** 3
**Presentation:** 3
**Contribution:** 2
**Rating:** 5
**Confidence:** 3

**Summary:**

This paper proposes a multimodal variational mixture-of-experts prior (MMVM) VAE to address challenges in learning representations of multimodal data. Instead of modeling the latent representation as a joint posterior, this work introduces a multimodal data-dependent prior to regulate the latent representation learned from the VAE for each modality. Experiments on multiple benchmarks demonstrate that the proposed method is promising.

**Strengths:**

(1) The proposed data-dependent prior is novel and very inspiring, providing a different direction for investigating multimodal representation learning.

(2) The experiments conducted on two medical datasets demonstrate that the proposed methods are beneficial and promising for real-world applications.

**Weaknesses:**

(1) There is no direct evaluation of the learned representation and the data-dependent prior to support the "soft-sharing of information between modalities" proposed by this work. For example, a naive investigation could examine the differences and overlaps among each $H(z_m|𝑋)$, and how similar each learned $q^{m}_{\phi} (z_m|x_m)$ is to their prior.

(2) In experiments, the exclusion of two multimodal VAE baselines, Daunhawer et al., 2020 and Palumbo et al., 2023, which are mentioned in the related work section, weakens the evaluations of the proposed work.

**Questions:**

(1) Regarding the metrics and classifiers mentioned in lines 233-235, are they the same as those defined in [1] and [2]? If yes, please cite them. If not, please provide a more detailed description.

[1] Shi, Y., Siddharth, N., Paige, B., Torr, P.: Variational mixture-of-experts autoencoders for multi-modal deep generative models. In: Advances in Neural Information Processing Systems (2019)

[2] I. Daunhawer, T. M. Sutter, K. Chin-Cheong, E. Palumbo, and J. E. Vogt. On the Limitations of Multimodal VAEs. International Conference on Learning Representations, 2022.

---

> ### Author Rebuttal · Authors · 2024-08-06
>
> Dear reviewer 6TPP,
>
> Thank you again for your insightful review and feedback. In the following, we will address and answer your comments and questions.
>
> ---
> > There is no direct evaluation of the learned representation and the data-dependent prior to support the "soft-sharing of information between modalities" proposed by this work. For example, a naive investigation could examine the differences and overlaps among each , and how similar each learned is to their prior.
>
> Thank you for this comment. We like the idea of directly comparing the latent representations of different modalities. In the table below, we compare the symmetric Kullback-Leibler (KL) divergence between the posterior approximations of the frontal and the lateral view of the MIMIC-CXR dataset to assess the differences between latent representations of different modalities.
> We compare the KL divergence of the independent VAEs to the KL divergence of the proposed MMVM VAE. Suppose the KL divergence of the MMVM VAE is lower than the KL divergence of the independent VAEs. In that case, it implies that the latent representations align and that the soft-sharing of information between modalities works.
>
> The table below shows that the symmetric KL divergence of the proposed MMVM is (much) lower than the KL divergence of the independent VAEs. We compute the symmetric KL divergence as
> $$KL = 0.5 \cdot KL[q(z | x_F) || q(z | x_L)] + 0.5 \cdot KL[q(z | x_L) || q(z | x_F)]$$
> using 10000 multimodal samples from the MIMIC-CXR dataset and across two seeds.
>
> | Method | KL divergence |
> | ------------- | -------------------- |
> | Independent | 26693.488 &plusmn; 11558.155 |
> | MMVM | 576.698 &plusmn; 48.882 |
>
>
> In the manuscript, we support the claim of the soft-sharing of information by evaluating conditional generation coherence. The difference in conditional generation coherence between the independent VAEs and the proposed MMVM is achieved through the different prior distribution between the two methods. Figures 3d-3f show that the proposed MMVM can conditionally generate coherent samples; hence, the soft-sharing of information between the modalities via the proposed prior distribution works. We will emphasize this more in the camera-ready version of our paper.
>
> ---
> > In experiments, the exclusion of two multimodal VAE baselines, Daunhawer et al., 2020 and Palumbo et al., 2023, which are mentioned in the related work section, weakens the evaluations of the proposed work.
>
> We show the results for the MMVAE+ in the rebuttal PDF. We excluded the results in the submitted manuscript because the more complicated MMVAE+ method could not outperform the aggregation-based baselines used in our manuscript.
>
> ---
> > Regarding the metrics and classifiers mentioned in lines 233-235, are they the same as those defined in [1] and [2]? If yes, please cite them. If not, please provide a more detailed description.
>
> Thank you for highlighting the missing references regarding the evaluation procedures. We use the same metrics as in [2, 3,6] and will add the corresponding references to the manuscript.
>
> ---
> Thank you again for your feedback. We welcome any additional suggestions, questions, or requests for information and encourage further discussions.

---

> > ### Author Response · Authors · 2024-08-09
> >
> > Dear reviewer 6TPP,
> >
> > please let us know if we could address your concerns with our rebuttal or if there are other question remaining.

---

> > ### Comment · Reviewer_6TPP · 2024-08-11
> >
> > I appreciate the authors' efforts in the detailed response. My concerns have been addressed, and I have increased my score.

---

> > > ### Author Response · Authors · 2024-08-11
> > >
> > > Dear reviewer 6TPP,
> > >
> > > thank your for appreciating our rebuttal effort and updating your score.
> > >
> > > Please let us know if other points remain unclear or could be improved.

---

### Official Review · Reviewer_eP7z · 2024-07-11

**Soundness:** 2
**Presentation:** 3
**Contribution:** 2
**Rating:** 5
**Confidence:** 3

**Summary:**

The paper introduces the multimodal variational mixture-of-experts prior (MMVM) VAE, a new approach to enhance representation learning for multimodal Variational Autoencoders (VAEs). The MMVM VAE overcomes the limitations of the alternatives that parameterize joint posteriors with restrictive assumptions by using a data-dependent prior that softly guides each modality’s latent representation towards a shared aggregate posterior, instead of imposing hard constraints. The experimental analysis on benchmarks and real datasets assesses the quality of the latent representations and preservation of modality-specific information.

**Strengths:**

- Effective modeling technique: the paper presents an effective modeling technique that enhances representation learning by incorporating inductive biases into the joint prior of the latent space in multimodal VAEs.
- The contributions are grounded by solid theoretical foundations.

**Weaknesses:**

- The experimental analysis presented in the main text is not fully convincing. While the evaluation is properly conducted on several datasets, the performance is assessed with only a single quantitative experiment in each case. The extensive analysis provided in the Appendix could be referenced in Section 5 to better support and strengthen the discussion.
- The paper would benefit from a more specific analysis of the limitations of related work in multimodal VAEs. This would enhance the motivation and context for the proposed approach.
- The paper does not explore potential solutions to mitigate the inability to perform unconditional sampling. Addressing this limitation would significantly improve the overall contribution and applicability of the work.

**Questions:**

**Major concerns**
- I found that Figures 2 and 3 are unclear in demonstrating the benefits of the proposed models due to the following reasons:
   - Showing several values of $\beta$ appears irrelevant. Fixing $\beta = 1$ would likely provide a fairer comparison and clearer results.
   - In representation learning literature, assessing meaningful representations is a challenging task with many proposed metrics and evaluations. Why do predicting metrics on linear/non-linear classifiers assess the quality of latent representations and coherence, respectively? Without references or explanations for these metrics, their validity in supporting the claims remains unclear.

**Minor concerns**

- I would move the “Encoder and Decoder” paragraph from Section 3 to the beginning of Section 4.

**Found typos**

- Line 98: “patient , .” $\rightarrow$ “patient.”.

**Limitations:**

The primary limitation of this work is its inability to perform unconditional sampling, which the authors acknowledge and discuss appropriately. However, while they mention several potential solutions to address this issue, they do not explore any of these alternatives. Efficient unconditional and conditional sampling are among the key advantages of VAEs. Incorporating a solution for unconditional sampling would significantly enhance the paper, allowing the model to generate unconditional samples and fully leverage the strengths of VAEs.

---

> ### Author Rebuttal · Authors · 2024-08-06
>
> Dear reviewer eP7z,
>
> Thank you again for your insightful review and feedback. In the following, we will address and answer your comments and questions.
>
> ---
> > The experimental analysis presented in the main text is not fully convincing. While the evaluation is properly conducted on several datasets, the performance is assessed with only a single quantitative experiment in each case. The extensive analysis provided in the Appendix could be referenced in Section 5 to better support and strengthen the discussion.
>
> Thank you for the comment. We like the idea of referencing the additional results provided in the appendix. We will include this reference in the camera-ready version of our manuscript. However, we are open to suggestions on what could be moved to the main text to make the core analysis more self-sufficient.
>
> We assess three different metrics in the main text of the paper: the quality of the learned latent representation, the quality of the conditional generation, and the reconstruction error. We evaluate the quality of the learned representations (Figures 3a-3c, 4a) using the classification performance of a linear classifier. The quality of the conditionally generated samples using pretrained nonlinear classifiers and the reconstruction error using the output negative log-probability of reconstructed samples. We describe the evaluation in more detail in Appendix B.2.
>
> ---
> > I found that Figures 2 and 3 are unclear in demonstrating the benefits of the proposed models due to the following reasons:
> Showing several values of beta appears irrelevant. Fixing beta=1 would likely provide a fairer comparison and clearer results.
>
> We disagree that comparing the different methods concerning multiple $\beta$ values is irrelevant. Figure 3 clearly shows that the proposed method can outperform previous works regarding the quality of latent representation and conditional generation without trading off reconstruction quality. The $\beta$-VAE paper [1] and earlier work on multimodal VAEs [2] already showed the regularizing effect of the $\beta$ parameter regarding trading off the learned representation against the reconstruction quality. Hence, evaluating different methods regarding multiple $\beta$-values approximates the (whole) range of representation-reconstruction trade-offs, thus showing the models’ robustness concerning $\beta$.
>
> Based on the feedback, we've created an improved plot version highlighting models with $\beta=1$ with a star (see rebuttal PDF).
> Why would a comparison based on a single $\beta$ value be more fair (in your opinion)? The rebuttal PDF highlights that not all baseline methods have optimal performance at $\beta=1.0$ and benefit from evaluating a set of different $\beta$ values.
>
> ---
> > In representation learning literature, assessing meaningful representations is a challenging task with many proposed metrics and evaluations. Why do predicting metrics on linear/non-linear classifiers assess the quality of latent representations and coherence, respectively? Without references or explanations for these metrics, their validity in supporting the claims remains unclear.
>
> Thank you for highlighting the missing references for our evaluation procedure. Using linear/non-linear classifiers is a standard procedure [3, 4, 5] for assessing learned representations, not only for VAEs. While linear classifiers should highlight the simplicity of the learned representations, non-linear classifiers should indicate the information present in the representations; see also [3, 4, 5]. We will add more references for the evaluation procedure to the camera-ready version of our manuscript.
>
> ---
> > I would move the “Encoder and Decoder” paragraph from Section 3 to the beginning of Section 4.
>
> Thank you for the suggestion. We like the idea of moving the mentioned paragraph to section 4. We will incorporate the changes in the camera-ready version of our paper.
>
> ---
> > The paper would benefit from a more specific analysis of the limitations of related work in multimodal VAEs. This would enhance the motivation and context for the proposed approach.
>
> Thank you for pointing this out. Adding a more specific discussion of the limitations of previous works will improve the paper. We will add the following paragraph to the camera-ready version of our paper:
>
> Daunhawer et al. [2022] show that aggregation-based multimodal VAEs cannot achieve the same generative quality as unimodal VAEs and struggle with learning meaningful representations depending on the relation between modalities. If we can predict one modality from another, mixture-of-experts-based posterior approximations perform best if only a single modality is given as input but struggle to leverage the complete information coming from all modalities. In contrast, product-of-experts-based approximations excel if the complete set of modalities is available. However, their difficulty with optimizing unimodal approximations lets them underperform if they only receive a single modality as input.
>
>
> ---
>
> > The paper does not explore potential solutions to mitigate the inability to perform unconditional sampling. Addressing this limitation would significantly improve the overall contribution and applicability of the work.
>
> We would like to highlight that the proposed method is not generative by design, as conditional generation of missing modalities and representation learning are arguably the main use cases of multimodal VAEs. We outline approaches for adding generative capabilities in future work. While presenting a solution for the inability to perform unconditional generation would further strengthen the paper, we see it beyond the scope of this paper. Introducing the MMVM prior distribution and presenting convincing experimental results is a substantial and strong contribution.
>
> ---
> Thank you again for your feedback. We welcome any additional suggestions, questions, or requests for information and encourage further discussions.

---

> > ### Author Response · Authors · 2024-08-09
> >
> > Dear reviewer eP7z,
> >
> > please let us know if we could address your concerns with our rebuttal or if there are other question remaining.

---

> > ### Comment · Reviewer_eP7z · 2024-08-09
> > **Rebuttal feedback**
> >
> > Dear authors,
> >
> > Thank you for addressing my concerns, specifically by strengthening the motivation, highlighting the results for $\beta=1$, adding more references to support the chosen metrics, and incorporating additional references to highlight the limitations of previous works. However, I still have a few concerns:
> >
> > - Regarding the experimental material in the Appendix, my suggestion was not to move a specific result to the main text but rather to incorporate references to the Appendix material within the main analysis and discussion. This could enhance the overall analysis and discussion, thereby strengthening the paper.
> >
> > - While I acknowledge that showing results for varying $\beta$ can be insightful, without sufficient motivation, it became difficult for me to visually understand the gains of the proposed method. Why does “Figure 3 clearly show […] without trading off reconstruction quality?” Perhaps I am missing something—could you please clarify this point?
> >
> > - Additionally, including more references to other works that have performed similar analyses on $\beta$-grids could better ground the evaluations.
> >
> >
> > I strongly disagree with your statement about the $\beta$-VAE. The phrase “trading off the learned representation against the reconstruction quality” is vague and imprecise, and it is repeated several times throughout the paper. I do not believe that “learned representation” is a well-defined measure in itself, especially considering the extensive work and effort from the community in this area.
> >
> > The $\beta$ factor primarily controls the latent overlap [1], and under certain circumstances [2], it can favor latent disentanglement, i.e., representations aligned with the underlying generative factors of variation in the data. This might be a desirable property or not, depending on the context—such as when accurate reconstruction is prioritized, as in [3].
> >
> > That said, I appreciate the inclusion of results for $\beta$=1, as it helped to more clearly highlight the contributions, and I thank the authors for making this adjustment.
> >
> > [1] Mathieu, E., Rainforth, T., Siddharth, N., & Teh, Y. W. (2019, May). Disentangling disentanglement in variational autoencoders. In International conference on machine learning (pp. 4402-4412). PMLR.
> >
> > [2] Burgess, C. P., Higgins, I., Pal, A., Matthey, L., Watters, N., Desjardins, G., & Lerchner, A. (2018). Understanding disentangling in $\beta $-VAE. arXiv preprint arXiv:1804.03599.
> >
> > [3] Rombach, R., Blattmann, A., Lorenz, D., Esser, P., & Ommer, B. (2022). High-resolution image synthesis with latent diffusion models. In Proceedings of the IEEE/CVF conference on computer vision and pattern recognition (pp. 10684-10695).

---

> > > ### Author Response · Authors · 2024-08-11
> > >
> > > Dear reviewer eP7z,
> > >
> > > thank you for your detailed reply. We appreciate your effort.
> > >
> > > We agree that the $\beta$-parameter has several effects on the learned model and its interpretation may vary depending on the context.
> > >
> > > From our experiments, please see the rebuttal PDF or the experiments in the submitted manuscript, we could clearly see that training several multimodal VAE implements with multiple $\beta$ values resulted in models that seem to trade off the classification accuracy and the reconstruction loss in different ways. We can see that high classification accuracy does not necessarily result in low reconstruction error on different datasets and that different methods seem to have different "$\beta$-dynamics. As you pointed out, the optimal model depends on the context and from our experiments, we can see that it is not always straightforward to choose the best model according to latent representation classification accuracy *and* reconstruction error.
> > >
> > > Showing this dynamic or trade off is the reason behind our evaluation with multiple $\beta$ values. However, we agree and thank you for it that highlight the models where $\beta=1$ improves our plots. We will formulate this more clearly and rephrase the mentioned sentence in the updated version our manuscript.
> > >
> > > Please let us know if additional points remain unclear.

---

> > > > ### Comment · Reviewer_eP7z · 2024-08-12
> > > >
> > > > Dear Authors,
> > > >
> > > > Thank you for your response. I understand that highlighting this dynamic or trade-off can be valuable. However, in my opinion, a more thorough discussion of the plot and a clearer visual justification of why the proposed method is superior are necessary. In the original submission, the discussion was somewhat unclear.
> > > >
> > > > With this point addressed, my concerns are mostly resolved, and I will recommend acceptance.

---

> > > > > ### Author Response · Authors · 2024-08-12
> > > > >
> > > > > Dear reviewer eP7z,
> > > > >
> > > > > Thank you for your response and your recommendation of acceptance.
> > > > >
> > > > > We agree that a more thorough discussion of our results is valuable and will improve the paper. We will add this to the camera-ready version of our paper.
> > > > > With that said, it would be highly appreciated if your recommendation of acceptance would be reflected in an increased score.
> > > > >
> > > > > Thanks again for the valuable discussion, we do not take it for granted and it is appreciated.

---

### Official Review · Reviewer_56cj · 2024-07-12

**Soundness:** 2
**Presentation:** 3
**Contribution:** 3
**Rating:** 6
**Confidence:** 4

**Summary:**

In this work, the authors proposed a new multimodal VAE architecture which, quickly summarized, both encoders and decoders factorized across modality latent variables, and the prior for each modality latent, $z_m$, is a mixture-of-experts composed of all the encoders. With this probabilistic model choice, the authors derive a number of interesting results, and show that this choice of prior is optimal to optimize the corresponding ELBO. Then, the authors empirically validate their findings by comparing their approach with existing methods in the PolyMNIST, Bimodal CelebA, and CUB datasets. Finally, the authors evaluate the quality of the latent samples on the Hippocampal Neural Activities dataset, showing that it can separate different odor types within the data.

**Strengths:**

- **S1.** The idea of the MMVM model is interesting, and it conceptually makes sense, overcoming the issue of combining the latent space that exists in current architectures. As so, it is significant to the community.
- **S2.** I've checked most derivations, and they look correct and sound.
- **S3.** The experiments show that the MMVM is at-good or better than existing works.

**Weaknesses:**

- **W1.** The experiments should compare with more modern approaches like [MMVAE+](https://openreview.net/forum?id=sdQGxouELX).
- **W2.** The presentation could be improved, there is a clear lack of derivations, and some details are hard to grasp (it took me several reads to fully understanding the proposed probabilistic model).
- **W3.** There are plenty of (to my eyes) over-statements and vague statements, depending on what the text requires. E.g.:
  - Line 68. "Require restrictive assumptions" Which ones? Why are they restrictive?
  - Lines 72-73. "cannot ... and require stronger assumptions". Why? Which ones?
  - Line 52: "thereby providing potential insights into the neural mechanisms underlying memory impairment"
  - Line 125: "As argued and discussed in this paper, such aggregation can be overly restrictive."
  - Line 183: "is directly related to pairwise similarities between posterior approximation distributions of positive pairs, similar to contrastive learning approaches but in a generative approach"
- **W4.** Similarly, conclusions from the experiments look over-optimistic to me. It is true that the proposed method work as- or better than the baselines, but the results are pretty close (see Fig 3) without standard deviations. Also, saying that the samples in Fig. 4d are separated seems like an over-stretch to me (I'd at least add transparency to the points).


PS: I'll happily increase my score once a newer method like MMVAE+ is included, which I find completely necessary given the scope of the paper.

**Questions:**

- **Q1.**  Why only independent and PoE were selected for the experiments in Table 1? I understand that MMVAE should have similar complexity to PoE.

**Limitations:**

Limitations are properly discussed.

---

> ### Author Rebuttal · Authors · 2024-08-06
>
> Dear reviewer 56cj,
>
>
> Thank you again for your insightful review and feedback. In the following, we will address and answer your comments and questions.
>
> ---
> > The experiments should compare with more modern approaches like MMVAE+.
>
> We show the results for the MMVAE+ in the rebuttal PDF. We excluded the results in the submitted manuscript because the more complicated MMVAE+ method could not outperform the (stronger) aggregation-based baselines used in our manuscript.
>
> ---
> > The presentation could be improved, there is a clear lack of derivations, and some details are hard to grasp (it took me several reads to fully understanding the proposed probabilistic model).
>
> Thank you for highlighting this. We will improve the presentation for the camera-ready version of our paper. In particular, please point us to specific sections that you would like to see improved; we appreciate the suggestions.
>
> ---
> > There are plenty of (to my eyes) over-statements and vague statements, depending on what the text requires. E.g.:
>
> Thank you for these remarks. We will formulate them more precisely in the camera-ready version of our paper.
>
> Regarding the difficulties of defining joint posterior approximation distribution: We explain in lines 44 and 45 why it is challenging to design good aggregation functions that fulfill all the required properties. Due to space constraints, we do not repeat the statements in section 2 (lines 68, 72-73) or section 3 (line 125). However, reiterating this could improve clarity, and we will include it in the camera-ready version.
>
> Regarding reducing the pairwise similarities: In the first paragraph of section 3, we describe in our dataset assumption that the different modalities of a multimodal sample describe the same underlying phenomenon, making them positive pairs using contrastive learning terminology. We derive in lines 178 and 179 why the regularizer in the proposed objective is equal to minimizing the Jensen-Shannon divergence between M distributions. Hence, we accurately describe what we mean by minimizing the distance between positive pairs. We will add more clarification to the camera-ready version of our manuscript.
>
> ---
> > Similarly, conclusions from the experiments look over-optimistic to me. It is true that the proposed method work as- or better than the baselines, but the results are pretty close (see Fig 3) without standard deviations. Also, saying that the samples in Fig. 4d are separated seems like an over-stretch to me (I'd at least add transparency to the points).
>
> Figure 3 shows a clear outperformance of our method compared to previous work. We highlight that the proposed MMVM VAE achieves a higher representation quality or lower reconstruction loss (or both). In the uploaded rebuttal PDF, we also show the outperformance compared to MMVAE+ on the translated PolyMNIST and CelebA datasets. Table 1 also shows that the proposed MMVM outperforms previous works (incl. a supervised method) on the challenging MIMIC-CXR dataset (please see the additional results in the rebuttal PDF).
> We acknowledge that the performances on the hippocampal neural activities dataset are more comparable across the different methods. We will revise that part of our conclusion accordingly. Thank you for highlighting this.
>
> ---
> > Why only independent and PoE were selected for the experiments in Table 1? I understand that MMVAE should have similar complexity to PoE.
>
> Although we did not state it explicitly, we reported the results for the PoE VAE to highlight the performance of aggregation-based methods, as it is the best-performing approach in the MIMIC-CXR experiment. However, we understand the reviewers' feedback and will include an updated table with the performance of all baseline methods in the camera-ready version of our manuscript. The updated table is available in the rebuttal PDF.
>
> ---
> > PS: I'll happily increase my score once a newer method like MMVAE+ is included, which I find completely necessary given the scope of the paper.
>
> We added the results of the MMVAE+ to the rebuttal PDF. In the submitted manuscript, we focus on comparing the newly introduced MMVM VAE to independent and aggregation-based VAEs. As the results of the MMVAE+ cannot outperform stronger aggregation-based multimodal VAEs, we did not include them in the evaluation. However, we are happy to include the MMVAE+ results in the camera-ready version of the paper.
>
> ---
> Thank you again for your feedback. We welcome any additional suggestions, questions, or requests for information and encourage further discussions.

---

> > ### Author Response · Authors · 2024-08-09
> >
> > Dear reviewer 56cj,
> >
> > please let us know if we could address your concerns with our rebuttal or if there are other question remaining.

---

> > > ### Comment · Reviewer_56cj · 2024-08-11
> > >
> > > Dear authors,
> > >
> > > Thank you for the detailed response. I am mostly satisfied with the answers provided to mine and the other reviews, and I will thus update my score to reflect this.
> > >
> > > As of the presentation, my main suggestion is to have clearly in a single spot of the main text a couple equations clearly stating the probabilistic model (i.e., likelihood, prior, and posterior).

---

> > > > ### Author Response · Authors · 2024-08-11
> > > >
> > > > Dear reviewer 56cj,
> > > >
> > > > thank you for appreciating our effort and updating your score.
> > > >
> > > > We will add a probabilistic model to the updated manuscript and improve the clarity of the derivations.
> > > >
> > > > Please let us know if there are more points that are unclear or could be improved.

---

### Author Rebuttal · Authors · 2024-08-06

Dear Reviewers,

We would like to thank all reviewers for providing comprehensive and valuable feedback. We particularly value the reviewers' recognition of our effective, interesting, and inspiring idea [**56cj**, **eP7z**, **6TPP**], the solid theoretical framework [**56cj**, **eP7z**], and the strong results from MMVM [**56cj**, **6TPP**].

Your thorough reviews have helped us identify potential areas for improvement and gain valuable perspectives that strengthen the contribution of our research. We have carefully addressed each of your comments. Based on the feedback [**56cj**, **eP7z**, **6TPP**], we include additional results and experimental evaluation in the rebuttal PDF, confirming the strong performance of the proposed MMVM VAE.

We include the results for MMVAE+ as an additional baseline [**56cj**, **6TPP**], which were excluded in the submitted manuscript as the MMVAE+ could not outperform other aggregation-based multimodal VAEs.
To address the feedback regarding our evaluation procedure using multiple $\beta$ values [**eP7z**], we specifically show the performance of the models, which were trained with $\beta=1$, and discuss their performance differences.
For the MIMIC-CXR experiment, we include more aggregation-based multimodal VAEs, which we did not include in the submitted manuscript [**56cj**] as we only showed the performance of the best-performing aggregation-based multimodal VAE.

We welcome any additional suggestions, questions, or requests for information and encourage further discussions.
Once again, we thank the reviewers for their time and effort in evaluating our work.

-----
## References

[1] Higgins et al., “Learning Basic Visual Concepts with a Constrained Variational Framework”, ICLR 2017

[2] Sutter et al., “Generalized Multimodal ELBO”, ICLR 2021

[3] Shi et al., “Variational Mixture-of-Experts Autoencoders for Multi-Modal Deep Generative Models“, Neurips 2019

[4] Oord et al., “Representation learning with contrastive predictive coding”, 2018

[5] Chen et al., “A Simple Framework for Contrastive Learning of Visual Representations”, 2020

[6] Daunhawer et al., “On the Limitations of Multimodal VAEs”, ICLR 2022

---

### Decision · Program_Chairs · 2024-09-25

**Decision:**

Accept (poster)

**Comment:**

The reviewers acknowledge the soundness and the technical novelty of the paper. The rebuttal has a positive impact on the final ratings and all the reviewers scored weak accept or higher. The ACs followed the reviewers' recommendation.